# The impact of phosphorus on projected Sub-Saharan Africa food security futures

Daniel Magnone [1] ✉, Vahid J. Niasar [2] ✉, Alexander F. Bouwman [3,4], Arthur H. W. Beusen [3,4], Sjoerd E. A. T. M. van der Zee[5,7] & Sheida Z. Sattari[6]

Sub-Saharan Africa must urgently improve food security. Phosphorus availability is one of the major barriers to this due to low historical agricultural use. Shared socioeconomic pathways (SSPs) indicate that only a sustainable (SSP1) or a fossil fuelled future (SSP5) can improve food security (in terms of price, availability, and risk of hunger) whilst nationalistic (SSP3) and unequal (SSP4) pathways worsen food security. Furthermore, sustainable SSP1 requires limited cropland expansion and low phosphorus use whilst the nationalistic SSP3 is as environmentally damaging as the fossil fuelled pathway. The middle of the road future (SSP2) maintains today's inadequate food security levels only by using approximately 440 million tonnes of phosphate rock. Whilst this is within the current global reserve estimates the market price alone for a commonly used fertiliser (DAP) would cost US\$ 130 ± 25 billion for agriculture over the period 2020 to 2050 and the farmgate price could be two to five times higher due to additional costs (e.g. transport, taxation etc.). Thus, to improve food security, economic growth within a sustainability context (SSP1) and the avoidance of nationalist ideology (SSP3) should be prioritised.

In 2015, the UN declared its ambition to "end hunger, achieve food security and improve nutrition and promote sustainable agriculture" by 2030 as its second sustainability development goal (SDG)[1]. At the time, 8.8% of the world's and 21% of Sub-Saharan Africa's population was malnourished[2]. The food security situation was even worse: food security is defined by the UN Food and Agricultural Organisation (FAO) as being "when all people, at all times, have physical and economic access to sufficient, safe and nutritious food that meets their dietary needs and food preferences for an active and healthy life"[3]. By this measure in 2015 a quarter of the world's and over 50% of Sub-Saharan Africa's population was food insecure—and as high as 90% for some countries[2].

Food security consists of four key elements: food availability, access to food, utilisation of food and stability of food production. Food availability is primarily limited by the amount of food grown. Historically, increases in demand have been met by increased agricultural production intensity but this method of production comes at a high ecological and environmental cost:[4] to achieve the second SDG food security must be met whilst minimising ecological and environmental damage. Access to food is the ability of people to access sufficient food: factors such as income and power-differentials all affect this, hence, it is more complicated than food availability to quantify. Income also affects how food is utilised and, in turn, the effect of food production on the environment[5]. This is because increases in wealth often result in changes of diet[4] associated with higher calorie intake and often resource intensive foods (e.g. meat and dairy). These resource intensive foods may induce ecological and environmental damages as well as new challenges to human health[6]. Food waste also affects access to food: the UN estimates 930 million tons of food are wasted each year and minimising food waste through improved storage and distribution would reduce some production requirements[7]. The stability of food production is affected by both environmental (e.g. weather, soil fertility, natural disaster) and human factors (e.g. war, governance etc.)[5].

[1]University of Lincoln, Lincoln, UK. [2]University of Manchester, Manchester, UK. [3]Utrecht University, Utrecht, Netherlands. [4]PBL Netherlands Environmental Assessment Agency, The Hague, Netherlands. [5]Wageningen University, Wageningen, Netherlands. [6]Origin Enterprises Digital Limited, Didcot, UK. [7]Deceased: Sjoerd E. A. T. M. van der Zee. ✉e-mail: dmagnone@lincoln.ac.uk; vahid.niasar@manchester.ac.uk

To increase food availability and ensure stability of production, intensification of agricultural production has been historically employed. There are multiple strands to this (e.g. mechanisation, crop-enhancement, etc.) but arguably irrigation and fertiliser have the largest environmental footprints. Still, the increased and efficient use of fertilisers for nutrient enhancement is an important aspect of intensification[8,9]. Nitrogen, phosphorus and potassium are the most yield limiting nutrients for plant growth and are commonly added as fertiliser[10]. Throughout the 20th century more economically advanced regions of the world (e.g. Europe, North America, China) have improved yields by application of fertilisers[11]. Today these regions could, in theory, rely on residual (or legacy) nutrients, especially phosphorus due to high adsorption in soil and low release rate, to sustain high yields but do not commonly do so. Residual nutrients are the excess nutrients left in soil following years of high application and are often high enough to maintain high current yields: this is known as the *hysteretic crop uptake* effect[12].

The FAO declares phosphorus to be the most agriculturally yield limiting nutrient[10], and its supply is crucial to food availability. This is because it is critical to plant growth and there are no known biochemical alternatives—hence its description as "life's bottleneck"[13,14]. There are an array of phosphate fertiliser products, the main being compound NPK fertilisers, diammonium phosphate (DAP, containing 46% $P_2O_5$), monoammonium phosphate (MAP, containing between 48 and 61% $P_2O_5$) and triple superphosphate (TSP, containing ~44% $P_2O_5$), however the primary source of phosphate within all of these is phosphate rock[15] which is a finite resource[16,17].

Globally, under business as usual, it is estimated that the phosphorus footprint of 2050 will be 1.5 times that of 2010 due to population increase and dietary change[8,18]. But despite concerns in recent decades[16,17] humanity is unlikely to run out of phosphate rock and instead peak phosphorus will be reached at the point at which demand exceeds the supply of high grade phosphate (analogous to the more well-known "peak oil" concept). This peak is predicted to occur between 2070 and 2080[13]. There are an estimated 65 billion tonnes of phosphate rock in global reserves (i.e. the amount known to be economically extractable at high enough quality for use) and an estimated 300 billion tonnes in global geological resources (i.e. estimated concentrations not necessarily economically extractable)[19].

Fertiliser use is increasing most rapidly in developing countries[20], therefore, both the amount and the price of phosphate are important to agricultural productivity: price was a limiting factor during the phosphate crisis of 2008 and is increasingly the case in 2022. Since 1960 the annual prices of both DAP and TSP have been strongly correlated with the phosphate rock price (both 0.88, $n = 60$, $p < 0.01$). Throughout this period, the mean price of a tonne of DAP has been $4.60 \pm 1.28$ times higher than the price of phosphate rock whilst a tonne of TSP has been $3.86 \pm 1.02$ higher. Only twice since 1965, in 2008 & 2011, has the TSP price exceeded DAP price. Between 1970 and 2005 phosphate fertiliser price was relatively stable with the DAP commodity price rising gradually from US$ 68.5 to 225 per tonne and TSP rising from US$ 45 to 200 per tonne[21]. For this period, DAP and TSP had respective mean prices of US$ $164 \pm 50$ and US$ $140 \pm 50$ per tonne (including a minor spike in 1975). In 2008 prices increased by 525 % and 630 %, respectively for DAP and TSP from those baselines to US$ 860 and US$ 880 per tonne, respectively. This was due to a range of short-term issues resulting in demand outstripping supply[13,16]. By the end of 2009 prices had reduced, although not to pre-spike limits, and the price during the 2010–2020 decade has been US$ $405 \pm 70$ and US$ $375 \pm 90$ per tonne (mostly decreasing) for DAP and TSP, respectively. This means that the last decade has had both the highest sustained phosphate price and greatest fluctuations of any decade since 1960. Recently, the phosphate price has peaked again increasing 250% in the year from April 2021 to April 2022[21].

The extent to which fertilisers, including phosphates, are used is heavily influenced by socio-economic and socio-technical factors[22,23], the trajectory of which may take several different pathways. The conceptual framework of Shared Socioeconomic Pathways (SSPs) has been developed to account for human behaviour in the field of environmental and climatic modelling[24,25]. It outlines five alternative plausible societal trends, which will have varying impacts on ecosystems and the economy: SSP1 represents sustainability and a greener future; SSP2 middle of the road; SSP3 regional rivalries (i.e. increased nationalism); SSP4 is a divided an unequal society and SSP5 represents fossil fuelled development[26,27]. In this paper nationalism, as referred to in SSP3, refers to O'Neil et al.'s[27] definition from the original description of SSP3, that is that "policies shift over time to become increasingly oriented toward national and regional security issues, including barriers to trade, particularly in agricultural markets". More details on the scenarios outlined in the SSPs can be found in the supplementary information.

These scenarios have been built into the Integrated Model to Assess the Global Environment (IMAGE 3.2)[28–30] which projects crop production, agricultural efficiency, land use change (including agricultural land use), food prices and trade for the 21st Century within the SSP framework on a spatially gridded scale. Such projections are based upon socio-economic and socio-technical factors such as population, gross domestic product (GDP), policies, technology, lifestyle (e.g. dietary change) and resources and physical factors such as accessibility, terrain and soil properties[28].

The challenge of increasing food security is greatest in Sub-Saharan Africa[4,31] which is further compounded due to widespread limited phosphorus use. Food security SSPs scenarios for the coming century have been mapped by a previous study using MAGNET—an agro-economic model coupled to IMAGE[5]. Total food demand is a function of multiple factors including the total number of calories required, total population, wealth and trade processes. The lowest Sub-Saharan Africa population growth is likely to be under sustainable SSP1 and fossil fuelled SSP5 both of which have 1.6 billion people by 2050[30]. The highest population growth is likely to be under the increased nationalism of SSP3 and inequality of SSP4 both of which have populations of 2 billion people projected by 2050[30]. The middle of the road SSP2 reaches 2050 with 1.8 billion people (Fig. 1A)[30].

Calorie intake is expected to increase in Sub-Saharan Africa in the coming century. In 2010, Sub-Saharan Africa had an average food availability of 2200 kcal capita$^{-1}$ day$^{-1}$ and by 2050, under sustainable SSP1 and fossil-fuelled SSP5, this is projected to have increased to 2810 and 2850 kcal capita$^{-1}$ day$^{-1}$, respectively. The other pathways still have increases but at a lesser rate: SSP2 (middle of the road): 2700 kcal capita$^{-1}$ day$^{-1}$, SSP3 (increased nationalism) 2440 kcal capita$^{-1}$ day$^{-1}$ and SSP4 (unequal) 2440 kcal capita$^{-1}$ day$^{-1}$. By contrast, Organisation for Economic Co-operation and Development (OECD) countries in 2010 were already at 3400 kcal capita$^{-1}$ day$^{-1}$ (Table 1)[5].

However, food security is also a function of the relative risks of hunger which is partially controlled by wealth and production prices. In 2010 about 200 million people in Sub-Saharan Africa were at risk of hunger. Only sustainable SSP1 and fossil fuelled SSP5 will have a reduction in the relative risks of hunger—both dropping to 50 million people (i.e. 25% of 2010 levels). In both cases, this is partially due to changes in wealth and food production prices. SSP1 will have substantially reduced the relative production price at 59% of 2010 levels and have a good increase in wealth (3.21 times with GDP per capita at US$ 5300). Fossil fuelled SSP5 will have a slight increase in production price (1.04 times) but the increase in GDP per-capita (4.48 times at US$ 7300) far exceeds this (Table 1). MAGNET-IMAGE modelling suggests that under fossil fuelled SSP5 by the end of the century most sectors of society will benefit with the purchasing power of unskilled agricultural and non-agricultural labourers increasing by at least fourfold by 2100[5]. Under nationalistic SSP3 and unequal SSP4 the risk of hunger is

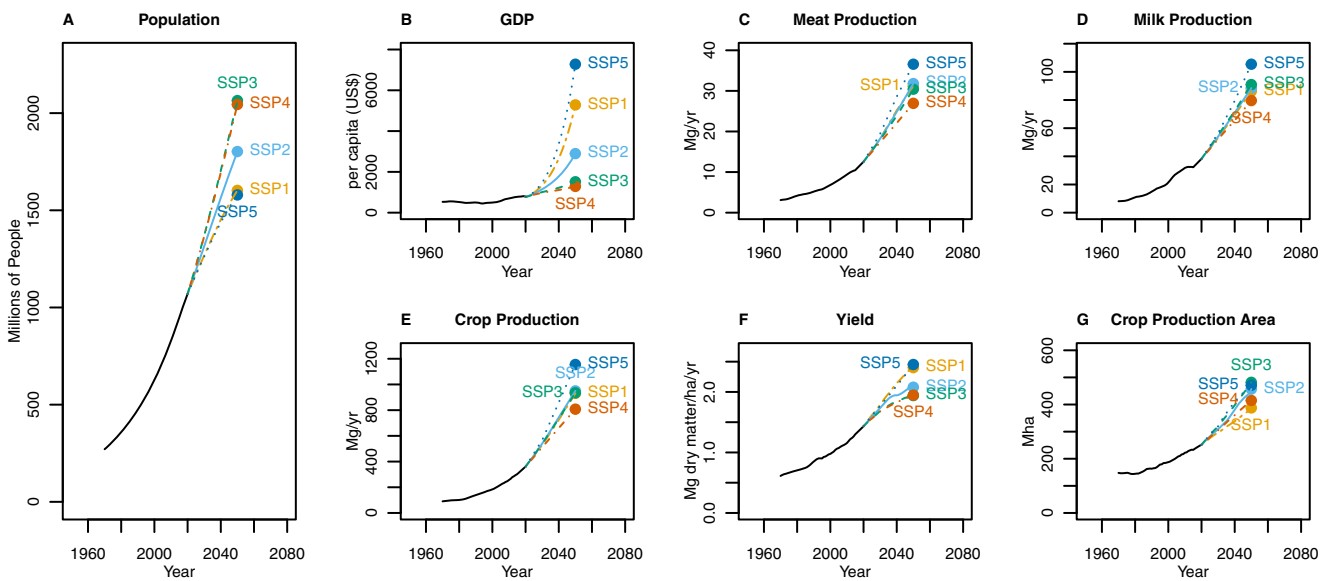

**Fig. 1 | Historical and future Sub-Saharan African for different Shared Socio-economic Pathways (SSPs) modelled using IMAGE 3.2.** (**A**) population growth (millions of people); **B** GDP (per capita US$); **C** Meat production (Mg yr⁻¹); **D** Milk production (Mg yr⁻¹); **E** Crop production (Mg yr⁻¹); **F** Yield (Mg dry matter ha⁻¹ yr⁻¹); and **G** Crop production areas (Mha) projected by IMAGE 3.2 for SSP1–5[30].

**Table 1 | Projected food security parameters for Sub-Saharan Africa at different SSPs for the 21st Century[5]**

| | GDP per capita | | Price production ratio ª (real market prices) | Availability | | Population at Risk of Hunger | | |
|---|---|---|---|---|---|---|---|---|
| | US$ | Ratio 2050 vs 2010 | 2050 vs 2010 | 2010 kcal capita⁻¹ day⁻¹ | Ratio: 2050 vs 2010 | Millions of people | Ratio: 2050 vs 2010 | Status |
| 2010 | 1650 | N/A | N/A | 2111 | N/A | 200 | N/A | N/A |
| SSP1 | 5300 | 3.21 | 0.59 | 2810 | 1.33 | 50 | 0.25 | Improved |
| SSP2 | 3000 | 1.81 | 0.96 | 2700 | 1.32 | 200 | 1.00 | No change |
| SSP3 | 2000 | 1.21 | 1.48 | 2440 | 1.16 | 300 | 1.50 | Worse |
| SSP4 | 2000 | 1.21 | 1.07 | 2440 | 1.16 | 300 | 1.50 | Worse |
| SSP5 | 7300 | 4.42 | 1.04 | 2850 | 1.35 | 50 | 0.25 | Improved |

ªRelative cost of food production in 2050 compared to 2010 where 2010 = 1.

projected to increase to over 300 million people. Under SSP3 this is partially because despite moderate increases in GDP per capita (1.16 times at US$ 2400) production prices will substantially increase to 1.48 times the 2010 levels due to the increase in land prices (Table 1)[5]. Furthermore, by the end of the century, inequality on these pathways is likely to have increased and unskilled agricultural and non-agricultural labourers are likely to have had a reduction in their purchasing power at <10% of 2010 levels; this increases the risk of hunger[5].

Overall, this indicates that only under fossil fuelled SSP1 and sustainable SSP5 will Sub-Saharan Africa become more food secure by the metrics of price, availability and risk of hunger. Under middle of the road SSP2 there is little change in food security but under the nationalism and inequality of SSP3 and SSP4 respectively, populations will become more food insecure (Table 1).

The second SDG emphasises the importance of food preferences within food security. Projected diets vary among the different pathways due to different levels of wealth. The domestic meat, milk and crop production would be highest under fossil-fuelled-high-GDP SSP5 and lowest under unequal-low-GDP SSP4 (Fig. 1C:E). Depending on the net trade of products, and the availability of productive land, meeting this demand will require closing the substantial continental ecological yield gap—the difference between potential and actual crop yields[32]. Since availability of good land is limiting in many regions, an increase in the intensity of cropping (e.g. irrigation, fertiliser use etc.) and/or the use of more agricultural lands are required[11,12,32,33] with much of the

yield limitations due to phosphorus deficiency in the soils[34]. Where nutrients are very low so-called "ecological intensification" can improve the ecological yield gap. Here, both organic matter is added to the soil and crop diversity is increased–including adding fertility crops to the rotation with[35].

In Sub-Saharan Africa the highest potential yields are in East Africa from Ethiopia to Zimbabwe and West Africa from Nigeria to Sierra Leone but many of these areas lack potential profitability[36]. The lack of potential profitability means there is a lack of financial incentives at the farm scale to close the so-called economic yield gap. The economic yield gap is the difference between current profit and maximum potential profit given the yield and is currently a quarter of the ecological yield gap[36]. However, at the national and continental scales the economic rewards of improved yields are great; agriculture currently makes up 20–50% of GDP of Sub-Saharan African nations and intensification of this sector is the most effective method of boosting national GDP and reducing poverty compared to other forms of industrial growth[37,38].

By 2050, the highest yields are likely to be under both the fossil fuelled SSP5 and the sustainable SSP1 (both 2.45 Mg dry matter ha⁻¹ yr⁻¹) whilst the lowest yields are under the unequal SSP4 (1.94 Mg dry matter ha⁻¹ yr⁻¹, Fig. 1F). Together this means that the production area is likely to be highest under SSP5 (470 Mha) followed by SSP3 (460 Mha), SSP2 (420 Mha), SSP4 (410 Mha) and lowest in SSP1 (380 Mha) (Fig. 1G). SSP5 was originally envisaged as "fossil fuelled growth"[27], however, we

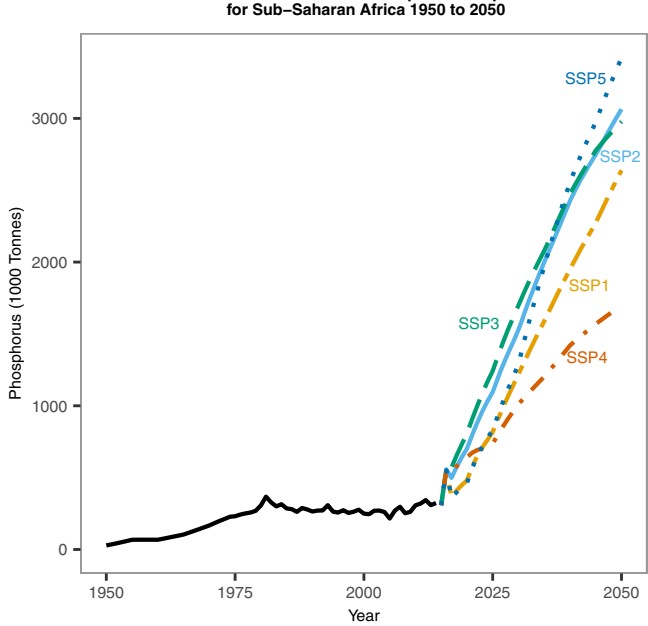

**Total Annual Elemental Phosphorus Required for Sub–Saharan Africa 1950 to 2050**

**Fig. 2 | Total elemental phosphorus required (1000 tonnes) in Sub-Saharan Africa from 1950 to 2050 (black line).** Using the model and input data the total elemental phosphorus required (1000 tonnes) has been estimated from 2020 to 2050 under different Share Socioeconomic Pathways (SSP1-SSP5)[27] within IMAGE3.2 framework[28].

broaden the term to mean environmentally exploitative (e.g. expanded agricultural lands, high water use, high greenhouse gas emissions) within the context of high GDP growth[26]. Whilst the volume and yield of crops vary under the different scenarios the crop production ratios (i.e. production of crop A compared to crop B) stay relatively constant in the IMAGE 3.2 scenarios. By 2050 "tropical roots and tubers" are projected to be the most abundant crop type in Sub-Saharan Africa in terms or fresh weight, followed by "maize", "tropical cereals" and "tropical oil crops" (see supplementary information).

The low nutrient applications throughout the 20th Century in Sub-Saharan Africa pose a significant challenge to economic and food security aspirations because the soils have been depleted of nutrients for decades[39]. At the continental scale, between 1961 and 1998, nitrogen, phosphate and potassium had severe depletion rates increasing by 225%, 233% and 256%, respectively[40], and the low crop yields in Sub-Saharan Africa have partially been attributed to this[11].

Phosphorus availability is a serious limiting factor in achieving the second SDG in many tropical countries[34]. Sub-Saharan Africa faces the greatest phosphorus challenge on the planet with one estimate suggesting that elemental phosphorus (i.e. pure phosphorus not in mineral or molecular form) application must increase fivefold from 4 kg ha$^{-1}$ in 2007 to 23 kg ha$^{-1}$ in 2050 to maintain food production—far more than any other continent[12]. The amount of phosphorus supplied from fertiliser varies across the continent: in 2005 52% of phosphorus in North Africa came from fertiliser, 29% in East Africa and 16% in Western Africa[41]. Furthermore, projected required inputs are not uniform across the continent, in part, because soil chemical properties of heavily weathered soils dramatically affect phosphorus availability to plants and hence the amount of phosphorus inputs required to grow crops on the short and long term[34,42–44].

In this work, we show that for Sub-Saharan Africa, two pathways—sustainable SSP1 and fossil fuelled SSP5—increase both food security (in terms of price, availability, and risk of hunger) and GDP by 2050. Of these sustainable SSP1 requires the lowest increase in phosphorus use and cropland expansion. By contrast, the nationalism of SSP3 and the

inequality of SSP4 both have worsening food security, furthermore, nationalistic SSP3 also induces high environmental damage with increased phosphorus use and cropland expansion. Crucially, just to maintain the current inadequate levels of food security (as predicted under middle of the road SSP2) a minimum of 440 million tonnes of phosphate rock will be required between the years 2020 and 2050 as well as cropland expansion; 74% of this phosphorus will be used in just 10 nations. The total cost of phosphate fertiliser (as DAP) will be US$ 130 ± 25 billion but the total farmgate price could be twice to five times this price due to additional costs (e.g. taxation, transport, etc.). This demonstrates that phosphorus is a critical limiter to a food secure African future but that economic growth within a sustainability context (SSP1) and the avoidance of nationalist ideology (SSP3) can help to deliver that better future.

## Results and discussion

### Projected phosphorus requirement in Sub-Saharan Africa under SSP1 to SSP5

We projected Sub-Saharan Africa's elemental phosphorus requirements using the IMAGE 3.2 database (described in the introduction) which, in turn, uses SSPs as an underlying framework[24,25,30]. IMAGE 3.2 projected crop production was coupled to the soil efficiency[28] using the Geochemical Dynamic Phosphorus Pool Simulator (GDPPS) which calculates efficiency based on the concentration of iron and aluminium oxides in the soil[45,46]. We projected SSP1–5 at a continental level and assessed specific countries under middle of the road SSP2 only.

Our results indicate that, regardless of SSP, Sub-Saharan Africa will require large increases in phosphorus application to maintain agricultural production in line with population growth and other socioeconomic factors (Fig. 2). In 2016, elemental phosphorus application via fertilisers was ~560,000 tonnes across the continent, however, by 2050, the lowest application of elemental phosphorus is likely to be under unequal SSP4 which requires 1.7 million tonnes, an increase of 310%. The highest application rates are in fossil fuelled SSP5 where total continental phosphorus application is projected to be 3.4 million tonnes, a 620% increase. The other SSPs (1, 2 and 3) follow similar trajectories to each other ending with values of 2.6 million tonnes (475%), 3.1 million tonnes (550%) and 3.2 million tonnes (530%) respectively.

Two pathways, sustainable SSP1 and fossil-fuelled SSP5, become more food secure by 2050 by the metrics of price, availability and risk of hunger (Table 1). Of these, fossil-fuelled SSP5 has the highest food production and hence the highest phosphorus use (Fig. 2) and the highest levels of GDP: its food security is a function of high purchasing power and high food availability (Table 1) from high crop, meat and milk production (Fig. 1C:E). The sustainable SSP1 on the other hand becomes more food secure (Table 1) but with far less phosphorus required (Fig. 2). This is because despite the high GDP, food production is much lower since efficient use of food provides greater access (Fig. 1B:E).

By contrast, the increased nationalism of SSP3 requires large amounts of phosphorus (Fig. 2) but has a worsening food security scenario (Table 1). Whilst the potential for food availability increases with increased food production (Fig. 1C:E), and associated increases in phosphorus (Fig. 2), the risk of food hunger increases in-part because food becomes less affordable (Table 1) with faltering GDP growth (Fig. 1B). The middle of the road SSP2 has similar food production (Fig. 1C:E) and phosphorus use as nationalistic SSP3 (Fig. 2) but improved GDP (Fig. 1B) and the scenario is no less food secure than today (Table 1).

The second SDG requires food security to be met with minimal environmental harm—part of which will relate to cropland expansion and phosphorus use—due to the inevitability of phosphorus run off and the related pollution[41,47]. Both the nationalism of SSP3 and the inequality of SSP4 become less food secure. Both have similarly high

rates of population growth and low rates of GDP (Fig. 1B), yet, SSP4 has the lowest phosphorus requirements whilst nationalistic SSP3 has relatively high phosphorus requirements (Fig. 2). This is caused by the expansion of croplands under nationalistic SSP3 which enables meat and milk production to increase at a faster rate than the unequal SSP4 despite similar GDP growth (Fig. 1B and G). Thus, the nationalism of nationalistic SSP3 will result in environmental damage from high phosphorus use and cropland expansion without improving food security.

Good land management, efficient resource use and technological change are modelled under the sustainable pathway of SSP1[27]. Here population growth is low and GDP comparatively high (although not as high as environmentally exploitative SSP5). Under sustainable SSP1, meat, milk and crop production increase along a middling pathway: greater than unequal SSP4 but less than environmentally exploitative SSP5 and similar to middle of the road SSP2 and the regional rivalries of SSP3 (Fig. 1C, D and E). However, sustainable SSP1 has a high yield increase (Fig. 1F), albeit slightly less than fossil-fuelled SSP5, and has the least expansion of croplands[30]. By targeting phosphorus efficiently high crop yields along with high meat and dairy production are sustained whilst limiting the amount of phosphorus used and the expansion of croplands. This highlights the necessity of targeted, efficient phosphorus with good land management use to prevent unnecessary expansion[8,9].

All together the analysis shows that a more food secure future, in terms of purchasing power, availability and risk of hunger, can be achieved with sustainable economic development as modelled in SSP1. The alternative approach to become more food secure is to enter an environmentally exploitative pathway (SSP5) with high cropland expansion and phosphorus use. By contrast the nationalism of SSP3 and unequal society of SSP4 become less food secure but SSP3 will have environmentally damaging cropland expansion and high levels of phosphorus requirements despite this. Finally, middle of the road SSP2 requires high amounts of phosphorus and cropland expansion without improving the levels of food security.

### Projected phosphorus requirements in different Sub-Saharan African countries under the middle of the road SSP2

We use the middle of the road scenario SSP2 to derive spatial and country specific information about the future phosphate projections[25–27]. Under this pathway food security retains its inadequate current levels (Table 1) so the phosphorus requirements mostly reflect population growth (Fig. 1).

Our results show that by 2020 elemental phosphorus uptake in Sub-Saharan Africa was mostly restricted to 7 kg ha$^{-1}$ or lower across most regions (Fig. 3A). Elemental phosphorus application had a maximum of about 7 kg ha$^{-1}$ but most of the continent had lower application rates (Fig. 3C). By 2050 we project these uptake rates will have increased to as high as 15 kg ha$^{-1}$ in Benin and Ghana, and ~12 kg ha$^{-1}$ over large parts of Cameroon, Gabon, Ethiopia, Nigeria, Republic of Congo, South Africa and Zambia (Fig. 3B). To support this, most of these nations will have to increase application rates to as high as 15 kg/ha (Fig. 3D).

We converted elemental phosphorus to both phosphate rock and DAP fertiliser (rock 70% BPL ≈ 32% $P_2O_5$; DAP = 46% $P_2O_5$) and projected that, under middle of the road SSP2, Sub-Saharan Africa will have consumed ~440 million tonnes of phosphate rock between 2020 and 2050. Over 74% of this will be required in just 10 countries: Nigeria 24%; Ethiopia 12%; South Africa 7%; Ghana 7%; D. R. Congo 6%; Côte d' Ivoire 5%; Cameroon 5%; Tanzania 3%; Benin 3% and Mali 3% (Fig. 3A). Between 2020 and 2050 we project the four highest phosphate consuming nations to consume 50% of the phosphate used.

Our results demonstrate that the minimum total projected Sub-Saharan African phosphate rock requirement between 2020 and 2050 is 440 million tonnes (Fig. 4A) which represents <1% of current known

global reserves (65 billion tonnes[19]). These results, however, assume that the system of phosphate rock mining to phosphorus application onto arable soils is 100% efficient, whereas a more accurate figure is likely to be 80%. It is unknown how efficiency will change over the next three decades[48]. Therefore, our phosphate rock estimate represents a minimum amount of phosphate required.

The increases in phosphate application are not evenly distributed. From a baseline of 2010 to 2020, we projected that phosphate requirement in Nigeria, Ethiopia, South Africa and Ghana will increase by 934%, 375%, 148% and 130%, respectively, over the 30 years to 2050. By 2050, under middle of the road SSP2, we project this consumption to be 3.19, 2.05, 0.64 and 0.92 million tonnes of fertiliser, respectively, if all is applied as DAP. For comparison, our modelled results from between the years 2010 and 2020 show that Nigeria, Ethiopia, South Africa and Ghana would have consumed an average of 0.29 ± 0.22, 0.45 ± 0.14, 0.44 ± 0.04 and 0.05 ± 0.08 million tonnes per year of fertiliser if all was applied as DAP (Fig. 4B,C). Were all applied as TSP the mass would be ~105% of these values whilst if they were applied as MAP the mass would be between 68% and 95% of the values depending on $P_2O_5$ concentration[36,49,50].

All countries we studied are required to increase their phosphate fertiliser market price expenditure. We project the total phosphate fertiliser market price cost to Sub-Saharan Africa between 2020 and 2050 to be US$ 130 ± 25 billion were all of this to be applied as DAP. For the four highest consuming countries, Nigeria, Ethiopia, South Africa and Ghana the market price expenditure in the year 2050 is projected to be US$ 1300 ± 230, US$ 815 ± 150, US$ 250 ± 45 and US$ 370 ± 70 million per year, respectively, up from a mean of US$ 105 ± 70, US$ 180 ± 50, US$ 180 ± 30 and US$ 20 ± 25 million US$ per year, respectively, between 2010 and 2020 (Fig. 4B). Were all of this applied as TSP the total cost would be ~90% of the DAP price given TSP has a lower price but also a lower $P_2O_5$ concentration[21].

To provide context to these figures we assessed phosphate fertiliser (as DAP) expenditure as a percentage of independently projected real GDP for each country (Table 2)[51]. Real GDP adjusts for inflation of goods and services by an economy for a given year and is expressed at base year prices. The results from South Africa and Ethiopia show that both countries have had historical peak expenditures in 1975 and 2008, respectively, but that this expenditure decreased and will continue to do so until 2050. On the other hand, we predict that Nigeria and Ghana have yet to reach their peak expenditure on fertiliser and will experience this peak around the year 2030 with Ghana experiencing a considerably higher peak than Nigeria (Fig. 4D).

Whilst these estimates provide insightful information about the phosphorus fertiliser use and cost in Sub-Saharan Africa in the coming decades, they should be considered approximations and have important caveats. The first caveat is that in practice a range of fertilisers exist, and much phosphate will be added as part of NPK fertilisers. We justify the choice of DAP since it is a widely used phosphate fertiliser and NPK has a variety of compositions[36,49,50] which prohibits the application of a single value required at the scale we are working. The second caveat is that this is not a calculation of the farmgate price, which usually exceeds the fertiliser price due to additional costs and can be between two to five times greater than the market price of fertilisers[50]. The final caveat is that our projections assume that the phosphate fertiliser stays within the price limits of the 2010–2020 decade (US$ 405 ± 70 per tonne of DAP[21]). This represents the decade of highest sustained and most fluctuating prices but, at the time of writing, the price is spiking almost twice this value. Nevertheless, as with previous spikes we expect this value to come down in the long term[21]. The extent to which farmers can maintain high yields can overcome short-term fertiliser price spikes by using less or no fertiliser is partly dictated by the amount of residual (or legacy) phosphorus and the so-called hysteretic crop uptake effect.

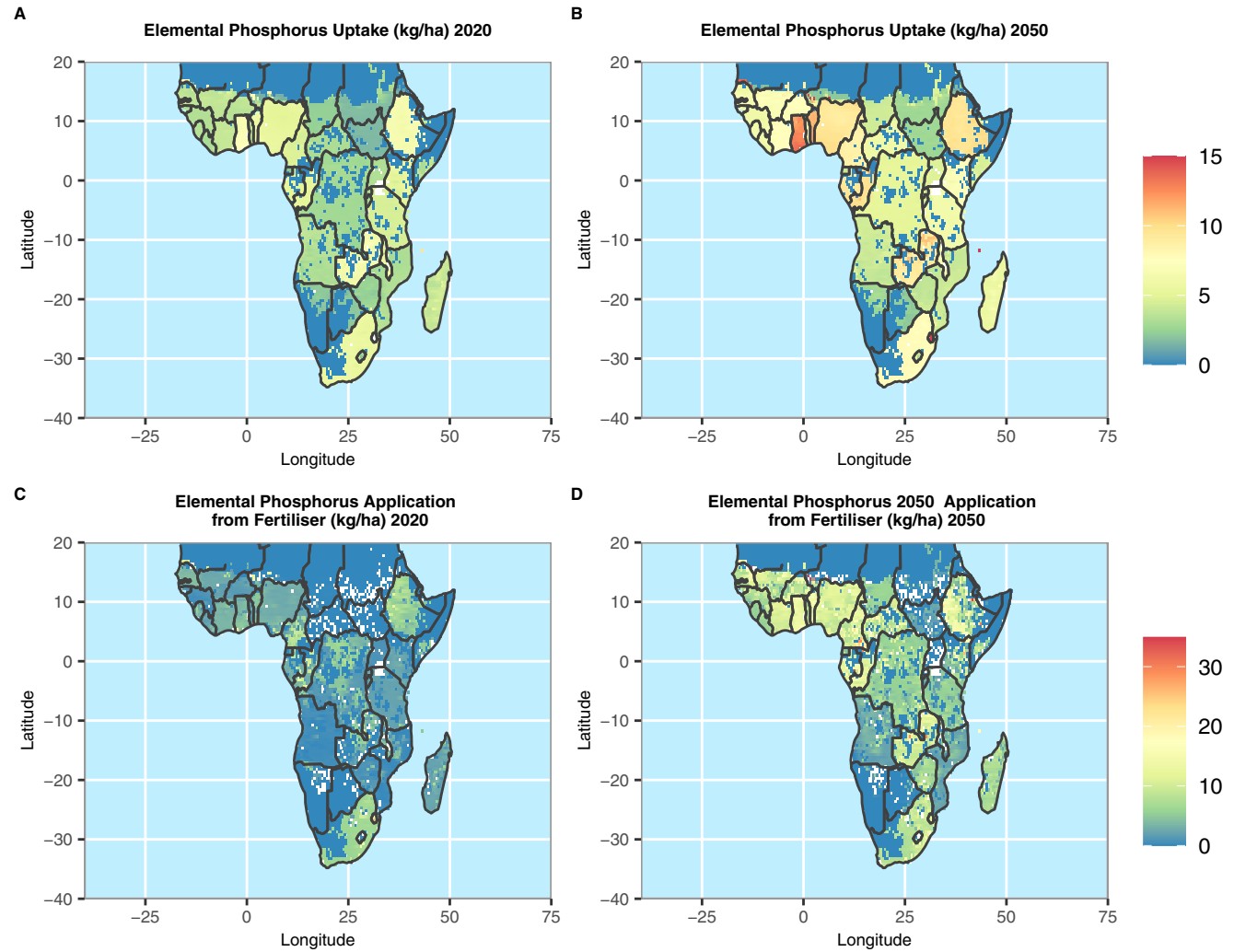

**Fig. 3 | Spatial distribution of annual phosphorus uptake and application rates (kg ha-1) in Sub-Saharan Africa for 2020 and 2050.** Spatial distribution of annual phosphorus uptake rates (kg ha⁻¹) in Sub-Saharan Africa for 2020 (**A**) and 2050 (**B**). Spatial distribution of annual elemental phosphorus application rates from fertilisers (kg ha⁻¹) for 2020 (**C**) and 2050 (**D**). All results were generated under the middle of the road SSP2 scenario. Country outlines made with Natural Earth vector files.

Under specific scenarios phosphorus uptake (and hence yield) can increase despite declining phosphorus input—this is called the hysteretic crop uptake effect. The hysteretic crop uptake effect is beneficial to regional agriculture since application rates become less important. At a continental level Western Europe has benefitted from this effect since 1970, Latin America and Asia are expected to benefit in the coming decades, however, Sub-Saharan Africa requires substantial increases in phosphate fertiliser application[12].

The hysteretic crop uptake effect occurs when part of the applied phosphorus is adsorbed by soil and is gradually released over time. The adsorption and release rates of phosphorus in soils depends on the soil chemistry, mineralogy and microbiological conditions. The difference in adsorption and release rates of phosphorus is the "hysteretic effect". To assess this effect, total applied phosphorus which, unlike the rest of this manuscript includes manure, is compared to phosphorus uptake. Total phosphorus is used since this is a physio-chemical process and both components will affect uptake.

Our results demonstrate that historically, Nigeria, South Africa, Democratic Republic of Congo and Cameroon have experienced the hysteretic crop uptake effect (Fig. 5). This is due to local variations in phosphorus flows within the soil which we discuss in detail in the supplementary information. However, no studied country is projected to experience the hysteretic effect under middle of the road SSP2. This is because crop uptake far exceeds the amount of residual nutrients. Most importantly, this means that just to maintain current standards of food security (as defined by price, availability and risk of hunger), all nations in this study will rely on phosphorus application as part of the intensification process.

## Methods
### Study design and assumptions
This study predicted future phosphorus requirements across Sub-Saharan Africa using data from the geochemical dynamic phosphorus pool simulator model (GDPPS)[46]. GDPPS uses a theory-driven geochemical fitting to overcome limitations induced in the conventional DPPS model where soils are highly weathered such as tropical locations[41,46]. Required yields were made according to the SSPs built into IMAGE 3.2 (see introduction for details)[24,25]. Approximation on phosphate expenditure was estimated using the mean DAP market price (≈46% P₂O₅) between 2010 and 2020 (US$ 405 ± 80 per Metric ton of DAP) this is broadly comparable to the mean TSP market price (≈44% P₂O₅) for the same period (US$ 373 ± 82 per Metric ton of TSP). This historic data is described in detail in the introduction to this paper.

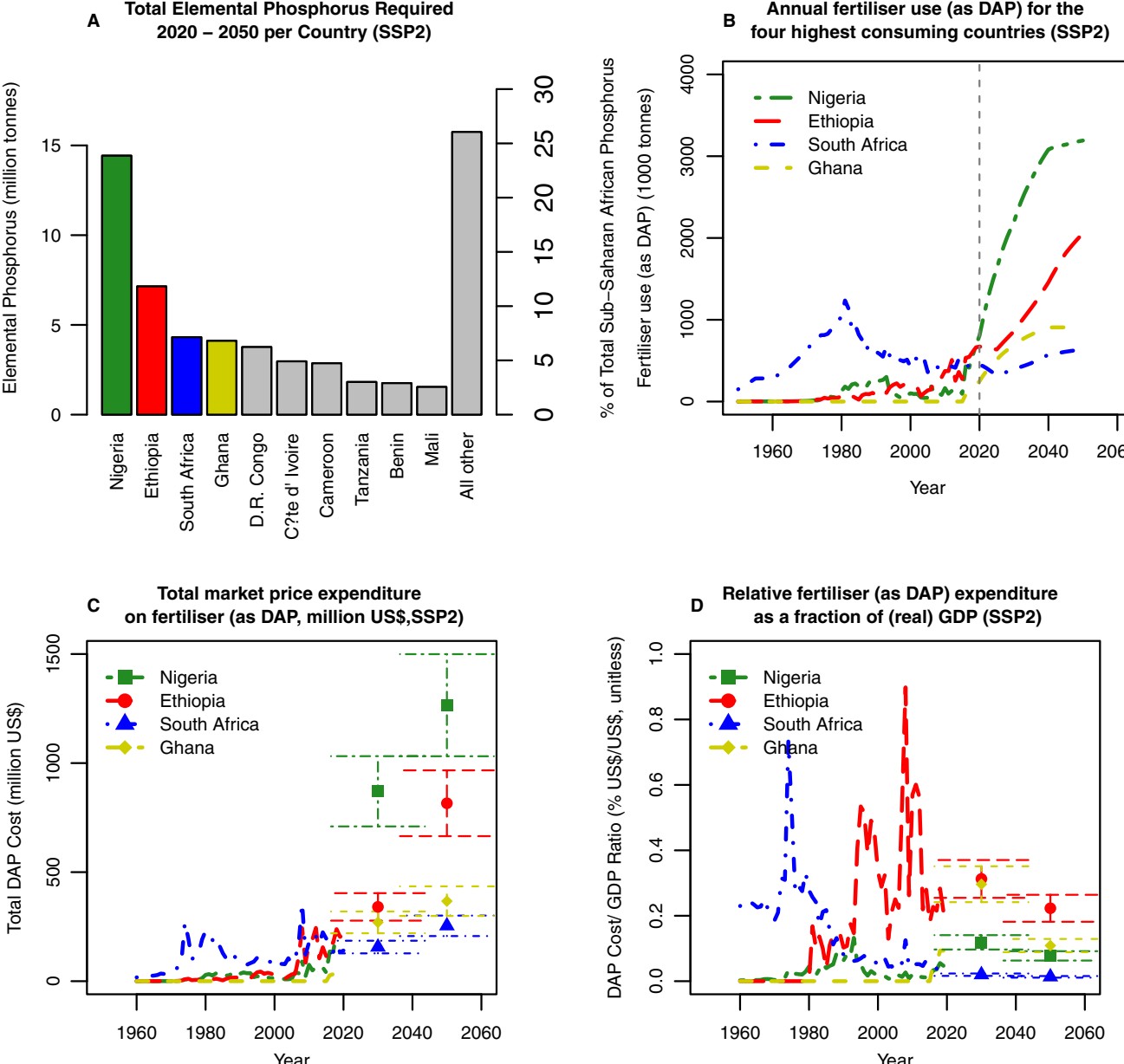

**Fig. 4 | Trajectory of phosphorus requirement and associated estimated costs in different Sub-Saharan African countries.** For SSP2 (**A**) Total phosphorus required (million tonnes) and percentage breakdown for Sub-Saharan Africa between 2020 and 2050; **B** Historic and projected DAP use (1000 tonnes) 1950–2050 in Nigeria, Ethiopia, South Africa and Ghana; and **C** historic and projected DAP costs (million US$) 1950–2050 in Nigeria, Ethiopia, South Africa and Ghana; and **D** historic and projected fertiliser expenditure (as DAP) relative to GDP

(USD $/USD $, unitless) from 1950 to 2050 in Nigeria, Ethiopia, South Africa and Ghana. Phosphorus, phosphate rock and DAP data are calculated by this project, historic DAP price and GDP are provided by the World Bank[21, 59] and projected GDP[51]. For figures **C** and **D** error bars represent 1 standard deviation propagated from the standard deviation of the price or expenditure, respectively, for the years 2010–2020.

The price of phosphate was compared to current and published projected real GDP for five countries (Nigeria, South Africa, Kenya, Ghana and Ethiopia; Table 2). The published data shown were projected using a model based on the Cobb-Douglas function[51].

**Table 2 | Projected Real GDP (US $) for different nations studied[51]**

| Country | Modelled Real GDP 2030 | Modelled Real GDP 2050 |
|---|---|---|
| Nigeria | 733 | 1636 |
| Ghana | 91 | 337 |
| South Africa | 791 | 1919 |
| Ethiopia | 109 | 366 |

## GDPPS model description

We developed GDPPS to calculate the phosphorus application as a function of the available phosphorus in the soil, the inputs and the phosphorus needed for crop uptake[46]. The independent key variable phosphate uptake is controlled by the amount of crop production as determined for an SSP within IMAGE 3.2—this is calculated at a gridded scale of 0.5° × 0.5°[41,46,52]. Phosphorus use efficiency in SSA has relatively high values and fluctuates over a small range (0.8–1) these do not affect our results since they are a function of our model[39].

In GDPPS, soil phosphate has a two-pool structure including a partially labile pool (labile forms) which is partially available to plants and a stable pool which is unavailable to plants (stable forms). Chemically, phosphate is retained in soil within a continuum of bonding

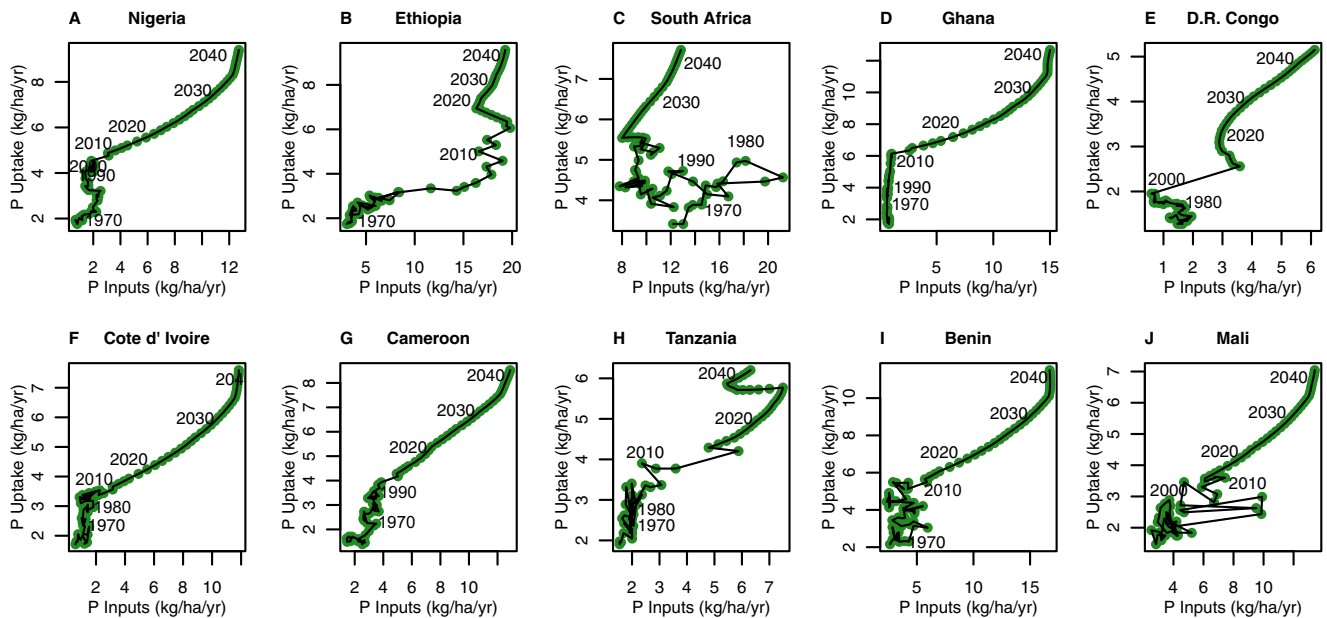

**Fig. 5 | Historical trend between the phosphorus application and uptake in different Sub-Saharan African countries.** Total phosphorus inputs (fertiliser and manure, kg ha⁻¹ yr⁻¹) vs total phosphorus uptake (kg ha⁻¹ yr⁻¹) for the period 1970–2050 for the 10 highest phosphorus consuming countries in Sub-Saharan Africa under middle of the road SSP2: (**A**) Nigeria, (**B**) Ethiopia, (**C**) South Africa, (**D**) Ghana, (**E**) Democratic Republic of Congo, (**F**) Cote d' Ivoire, (**G**) Cameroon, (**H**) Tanzania, (**I**) Benin, and (**J**) Mali.

energies with varying degrees of reversibility, ranging from labile forms (partially available to plants) to more stable forms (less available to the plant). In soils with neutral to acidic conditions, the transfer between the different forms is controlled by the concentration of iron and aluminium oxides in soils[45,46]. Our model describes this transfer between the two pools as controlled by the concentration of weathered iron and aluminium oxides[46]. A diagram is provided in the Supplementary Information for additional clarity of the model structure.

The governing equations are provided below (Eqs. 1 and 2). A temporal change of the labile pool size ($dL/dt$) is due to input fluxes (kg ha⁻¹ yr⁻¹) from litter ($f_{lit}$), fertiliser ($f_{Fert}$), manure ($f_{Man}$), weathering ($f_{wt}$), fresh soil ($f_{fsL}$), dissolution of the stable pool ($S.r_s$) and output fluxes of runoff ($f_{RL}$) and crop uptake ($f_c$)[12,52]. Transfer between the labile and stable pools is controlled by Kinetic Soil P model (KINS-P) which is derived from the van der Zee and van Riemsdijk model[44,53]. In KINS-P the transfer is controlled by concentration of oxalate Fe and Al ($M$, kg ha⁻¹) multiplied by an activity constant ($k$) divided by the labile pool ($L$, kg ha⁻¹).

$$\frac{dL}{dt} = \frac{f_{lit} + f_{Fert} + f_{Man} + f_{wt} + f_{fs_L} - f_{R_L} - f_c + S \bullet r_s}{1 - \frac{k \bullet M}{L}} \quad (1)$$

The change in stable pool size ($dS/dt$) is controlled by the balance of input fluxes (kg ha⁻¹ yr⁻¹) from atmospheric deposition ($f_{At}$), fresh soil ($f_{fsS}$) and outputs of runoff ($f_{Rs}$), dissolution of the stable pool ($S.r_s$) plus KINS-P multiplied by the temporal change of the labile pool size.

$$\frac{dS}{dt} = f_{At} + f_{fs_s} - f_{R_s} - S \bullet r_s + \frac{k \bullet M}{L} \bullet \frac{dL}{dt} \quad (2)$$

## Model data
The input and output data was from the IMAGE 3.2 project[54]. Virgin pool sizes were provided by Yang et al.[55], using 0.5° × 0.5° gridded scale. Values for M were provided by ISRIC at a gridded scale of 250 m[56]. Hengl et al.[56] produced these data using soil samples from about 59,000 locations and remote sensing co-variates to cover the whole continent. Samples for extractable Fe has good coverage in West Africa between Nigeria and Ghana as well as in east Africa from Tanzania through Ethiopia with no samples in the Congo basin. Values from M were maintained constant throughout the model simulations (see supplementary information for details on M stability). The activity constant, $k$, was determined by this study for each individual grid cell from the virgin soils conditions assuming that $a \approx 1$[44] and that soils had been in equilibrium for 400 years[57,58]. The dissolution rate constant, $r_s$, was determined individually for all grid cells from virgin soils assuming no change of the pools and that only the natural forcings (e.g. deposition, weathering, runoff) and no anthropogenic (e.g. fertiliser, manure) acted on the system for the year 1900[46].

## Model validation
During the development of GDPPS, the model was validated by comparing the modelled soil phosphorus pool sizes to measured phosphorus pools. Measured dataset consisted of 17,160 georeferenced soil profiles collected between 1950 and 2000 from across the continent. Validity was assessed using a lack of fit test, root-mean-square error, and Wilmot's index of agreement. The validation was extensive and published as open access work[46].

## Terminology

- *Phosphorus* = elemental phosphorus (P);

- *Elemental Phosphorus* = the amount of pure phosphorus (e.g. the amount uptake by a crop).

- *Phosphate* = phosphorus in oxidised form either as the dissolved phase in soil $PO_4^{2-}$ or mineral phase in rock $P_2O_5$ (see phosphate rock);

- *Phosphate rock* = The sedimentary, non-detrital rock used as the source for phosphorus fertiliser and a key commodity. We assume it to be 70% BPL which is approximately equivalent to 350 g of $P_2O_5$ per kg of rock.

- *Phosphate fertiliser* = fertiliser made from phosphate rock and commonly in the form of NPK fertilisers, diammonium phosphate (DAP, containing 46% $P_2O_5$), monoammonium phosphate (MAP, contain between 48 and 61% $P_2O_5$) and triple superphosphate (TSP, containing ≈44% $P_2O_5$).

- *Geological (Mining/Mineral) Resource* = the total amount of resource in the earth.

- *Geological (Mining/ Mineral) Reserve* = the amount of resource economically extractable – may change with economic and technical changes.

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

## Acknowledgements

This research was funded by an EPSRC Global Challenges Research grant (EP/111676), UMRI grant awarded to V.J.N., and an N8 Agrifood grant awarded to V.J.N. and D.M. Data used for this study were provided by the Integrated Model to Assess the Global Environment (IMAGE). A.F.B and A.H.W.B received support from PBL Netherlands Environmental Assessment Agency through in-kind contributions to The New Delta 2014 ALW project no. 869.15.015 and no. 869.15.014. We thank Dr. Eric Ruto (University of Lincoln) and Professor Gary Bosworth (Northumbria University) for economic advice and feedback during the drafting of this paper. Finally, we are very grateful to Professor Hans van Meijl (Wageningen University) who provided guidance on food security during the revisions of this paper.

## Author contributions

D.M.—Primary paper drafting, GDPPS theoretical development and concept development; V.J.N.—Research concept development and GDPPS theoretical development; A.F.B.—database development; A.H.W.B.—model development; S.v.d.Z.—GDPPS theoretical development; and S.Z.S.—concept development and advice on soil-phosphorus model. All authors have contributed to the writing and refining of this paper and the development of ideas within it (D.M., V.J.N., A.F.B., A.H.W.B., S.v.d.Z. and S.Z.S).

## Competing interests

The authors declare no competing interests.
