## [Peer Review File · Nature Communications]

REVIEWER COMMENTS

Reviewer #1 (Remarks to the Author):

This manuscript quantifies the amount of phosphate rock required by 10 nations in Sub-Saharan Africa (SSA) to sustainably intensify agriculture by 2050 under business-as-usual scenario, and, these nations' relative expenditure on phosphate rock. It typifies these countries as following one of three economic phosphorus pathways, one of which is a 'financial phosphorus bottleneck', highlighting the scale of the challenges facing SSA.

Optimising sustainable phosphorus use in SSA is critical to the continent's food security and the viability of their food systems. The issues addressed by the manuscript are important, and much work is required in the field at both the macro and micro level to better understand the nature and implications of phosphorus-specific challenges in SSA for long-term food security. Few studies have addressed the macro issues, which this manuscript does. The manuscript is novel in the authors' linking of their macro analysis in this manuscript to their recently published geochemical dynamic phosphorus pool simulator model (GDPPS) (Magnone et al 2019). Some of the authors are certainly experts in the field of soil phosphorus, and in their previous 2019 paper they highlighted a) the important implications of soil chemistry on phosphorus use efficiency and local and regional scales, and b) that the heterogenous nature of SSA's soil chemistry has not been factored into key influential models used to predict phosphorus fertiliser use on the continent.

There are however some significant limitations of the manuscript in its' current form. These mainly pertain to fundamental (non-soil) assumptions around food security, phosphorus and policy, and the transparency of these, as outlined below.

First, the study only models one single future scenario, rather than multiple. Why only use a business as usual scenario? Given the uncertainty and complexity surrounding future food security and land use, multiple scenarios would seem more appropriate.

Second, this business as usual scenario is based on the Shared Socioeconomic Pathways (SSP2). Why use SSPs rather than other future scenarios? The SSPs were designed to inform the analysis of climate impacts, vulnerability mitigation and adaptation. Whilst they do include land-use, achieving food security and sustainable food systems goes far beyond land-use issues. There is therefore a key missing link in this manuscript to the key current food security narrative. The global food security discourse is shifting beyond 'sustainable intensification' to a critical focus also on 'healthy diets' and 'sustainable food systems'. For example, the EAT-Lancet Commission of 37 organisations developed the "planetary health diet" (published in The Lancet). Transitioning to healthy diets within planetary boundary limits by 2050 will require substantial dietary shifts. Business as usual agricultural production systems will no longer be an option for the future. E.g. "Strategy two: reorient agricultural priorities from producing large quantities of food to producing healthy food". This means there is a need to transform food systems and grow what we need for healthy diets, rather than just making the current fundamentally flawed food system more 'efficient' through intensification. Sustainable intensification is important, but it alone cannot lead to food security and sustainable food systems. E.g. in SSA, a significant increase in consumption of vegetable, fruits, whole grains, nuts, and dramatic decrease in starchy vegetables, needs to take place for healthy diets, rather than improving the efficiency of the current crops under production. What would be the phosphorus demand in SSA countries under these production systems? EAT-Lancet has indeed answered part of this question, though the authors of this Nature manuscript could improve it with their soil model (in addition to the EAT-Lancet Commission refs - Willett et al and Springmann et al, see also the World Resources Institute and CGIAR's International Food Policy Research Institute, references below – Ranganathan et al and Robinson et al)

At least some (perhaps all) of this manuscript's authors are very much experts in their field of soil modelling and phosphorus, and it would be a tremendous contribution to new knowledge to model more relevant sustainable food scenarios. The EAT-Lancet Commission's work could be improved/built-upon to show what are the implications of shifting healthy diets (in a low food waste etc scenario) on phosphorus use and expenditure in SSA? To my knowledge, EAT-Lancet Commission did model the phosphorus implications of both BAU and healthy diets in 2050, but

perhaps did not include the soil phosphorus complexities that the Nature manuscript authors highlight. It may be that phosphorus demand needs to increase even more than using the SSP2 scenario. This would be valuable new knowledge with important implications. It would be interesting (and arguably necessary) to compare the manuscript authors BAU scenarios for 2050 with the EAT-Lancet projections for 2050.

A third important assumption behind this manuscript which needs to be clarified, and possibly revised, relates to phosphate rock concentration. The authors assume the concentration of phosphate rock is constant to 2050. However we know that as a finite resource, the average concentration of phosphate rock is and will continue to decline over the long-term as the high-grade phosphate rock is mined out (e.g. IFDC 2010). This should be included in the model, as it can significantly affect quantities and cost. The authors also use a surprisingly high average phosphate rock concentration of 35% P₂O₅ (350g of P₂O₅ per kg of rock), noted in lines 299-301 (p11). There are few existing deposits with this high grade, and certainly in the future this will decline. [further, this looks like a slight calculation error, as P₂O₅ = 0.4576 BPL, so the author's assumption of 70% BPL is actually equal to 32% P₂O₅, not 35% P₂O₅. This may also distort the results a little].

Fourth, and related to phosphate concentration, it appears the authors have chosen a static price for phosphate rock to 2050. Phosphate rock mining and processing costs will almost certainly increase over the long term, as new mine investments must be made, processing lower-grade phosphate rock in harder to reach locations and containing higher impurities (IFDC 2010). There are many uncertainties and debates around the longevity of phosphate rock, but the decline in concentration and increase in costs is not disputed. The cost per tonne of elemental phosphorus will increase, hence price, and this should therefore be reflected in the analysis presented here, rather than assuming costs will be stable through to 2050. It will be interesting to see how sensitive the authors' analysis of economic pathways are to a long-term price increase in phosphate.

Fifth, still on the topic of costs, the study looks at SSA nations expenditure of phosphate rock – but what about expenditure of phosphate fertilisers? (including fertiliser blends that include P among other nutrients?). Some African countries import finished fertiliser products (and re-blend to suit local conditions). Phosphate fertiliser costs are obviously also critical to this study. How have these been accounted for? What are the different phosphorus production and import profiles of the 10 SSA nations studied? Is most phosphorus demand in these countries met by imported mixed fertilisers, supplemented with a small amount of domestic phosphate rock production, and a small amount of imported phosphate rock imports? South Africa may be an anomaly as a large phosphate rock producing country. The point is, how do these different sources of phosphorus in these SSA countries affect the price and hence the results of the analysis?

Sixth, continuing on cost assumptions, how have the authors accounted for 'farm gate' cost of phosphate fertilisers? We know that farm-gate prices in land-locked African countries can be substantially higher than other parts of the world (e.g. 2-5 times higher due to transport, port duties, taxes, etc, see IFDC 2007). If the authors are using VCR to estimate affordability and "the extent to which fertiliser, including phosphate, is actually used on farmers" (line 60), then farm gate prices would be important to use. The costs in Figure 1 don't appear to consider farm gate prices? Further, farm gate prices would vary from country to country, so Figure 1B graph would not be able to use the same bars to match the different Y Axes of tons and cost.

Finally, the authors briefly touch on policy implications in lines 149-159. In the manuscript's current form, these important implications are not sufficiently explained/discussed, and they read as rather simplistic and somewhat inappropriate. E.g. the authors suggest that the only way forward for countries on the 'challenging' pathway are one of 2 options: either improve MVCR (e.g. through subsidies or transport improvements), "or else, the countries will have to supplement low yields with high rates of imported food" (lines 157-158). There is a whole myriad of other possible policy options that are not discussed here, possibly because the modelled assumptions themselves are so narrow, or due to the authors areas of expertise lying elsewhere. These policy implications should therefore be attended to in more expert and nuanced detail, or perhaps removed.

The authors are clearly experts in agricultural soil nutrient dynamics, as indicated by the level of attention given to these aspects (such as the entire Supplementary Information). However there are also many complexities, nuances and uncertainties in the fields of food security, phosphorus economics and policy implications, some of which have been highlighted above and many of which have not been addressed appropriately in this manuscript. The cost assumptions appear to oversimplify a very complex and dynamic situation, for example. The study would therefore at minimum need to demonstrate the sensitivity of the analysis to the variations and uncertainties indicated above, and address the specific comments below. The analysis would benefit from using multiple scenarios, and, scenarios that go beyond sustainable intensification to address the current food systems discussions around what would it take to achieve healthy diets and hence food security. The study needs to link to related research, such as the EAT-Lancet Commissions work, which does model phosphorus scenarios to 2050 (see also their Supplementary Material linked below). If the authors do not want to get into the food security debate, then they should significantly modify and qualify the stated intentions and findings of this study to refer to improving agricultural productivity, NOT food security.

References:

Prof Walter Willett, MD, Prof Johan Rockström, PhD, Brent Loken, PhD, Marco Springmann, PhD, Prof Tim Lang, PhD, Sonja Vermeulen, PhD, et al. (2019), 'Food in the Anthropocene: the EAT-Lancet Commission on healthy diets from sustainable food systems' THE LANCET COMMISSIONS| VOLUME 393, ISSUE 10170, P447-492, FEBRUARY 02, 2019
<https://www.thelancet.com/commissions/EAT>

Willett et al Supplementary material: [https://www.thelancet.com/journals/lancet/article/PIIS0140-6736\(18\)31788-4/fulltext#supplementaryMaterial](https://www.thelancet.com/journals/lancet/article/PIIS0140-6736(18)31788-4/fulltext#supplementaryMaterial)

Janet Ranganathan, Daniel Vennard, Richard Waite, Brian Lipinski et al; (2016), 'Shifting Diets for a Sustainable Food Future', Creating a Sustainable Food Future, Installment Eleven. World Resources Institute https://files.wri.org/s3fs-public/Shifting_Diets_for_a_Sustainable_Food_Future_1.pdf

IFDC (2007). 'Fertilizer supply and costs in Africa', prepared by Chemonics International Inc. and the International Center for Soil Fertility and Agricultural Development.

IFDC (2010). 'World phosphate rock reserves and resources'. Tech. Bull. T-75. International Fertilizer Development Centre (IFDC). Washington.

SHERMAN ROBINSON, DANIEL MASON D'CROZ, SHAHNILA ISLAM, TIMOTHY B. SULSER, RICHARD D. ROBERTSON, TINGJU ZHU, ARTHUR GUENEAU, GAUTHIER PITOIS, MARK W. ROSEGRANT (2015), The International Model for Policy Analysis of Agricultural Commodities and Trade (IMPACT): Model description for version 3, IFPRI DISCUSSION PAPER. 2015
<https://www.ifpri.org/publication/international-model-policy-analysis-agricultural-commodities-and-trade-impact-model-0>

Springmann, Marco; Clark, Michael; Mason-D'Croz, Daniel; Wiebe, Keith; Bodirsky, Benjamin Leon; Lassaletta, Luis; de Vries, Wim (2018) 'Options for keeping the food system within environmental limits. (Technical report)' Nature, 2018, Vol.562(7728), p.519

Specific comments:

Line 27-28: There are a few value-laden bold statements made in the manuscript, which need to be tempered and qualified, given the complexities and uncertainties highlighted above. E.g. "SSA requires a 2.5-fold increase in food production during the 21st century, in order to become food secure". This prediction is very much under the 'current' business as usual production paradigm. Even if production and productivity increase, there is no guarantee SSA food consumers have financial and physical access to healthy diverse foods essential for food security (not to mention the current production paradigm has ~40% food losses high up in the value chain). We know

business as usual is not an option and food systems need to dramatically transform. This has serious implications for phosphorus demand, as discussed in general comments.

Line 70-76: Many of these VCRs are from different sources/references. How comparable are they? Is there likely to be wide variation between them due to the reference year? Or due to different assumptions and data sources/reliability? You have noted in the paragraph below that VCRs can change dramatically year to year.

Line 73-76: This list in brackets reads a little awkward in the text. Perhaps put in a simple table for ease of readability, including year of VCR for transparency.

Line 115: Given these are future projections and scenarios, best to say "Our results suggest that...", or "These results indicate that ...are likely to have very different phosphorus pathways".

Line 122-125 (FIGURE 1): Figure 1 appears to be in a very draft form with quite a few errors:

- > First, there are 3 graphs (A, B, C) but only A and B are mentioned in the figure heading. In the Figure heading, (A) should refer to (B), and (B) should refer to (C). The first sentence in line 123 should refer to (A).
- > Second, Graph A shows 2020-2050 while Graphs B and C show 2018, 2030, 2050. Obviously it would be better if these date ranges were consistent.
- > Third, Graphs B and C should indicate whether 2018 data is actual or modelled, to get a sense of comparability with 2030 and 2050.
- > Fourth, In Graph A, 2 of the bars exceed the Y axis scale, and in Graph C, 2 of the bars in 2030 exceed the Y Axis scale. The scale should obviously be extended.

Line 300-301: "We assume it to be 70% BPL which is approximately equivalent to 350g of P₂O₅ per kg of rock". This is slightly incorrect. I assume the authors meant 320g of P₂O₅. They have noted the correct calculation on line 235 "70% BPL ~ 32% P₂O₅".

Reviewer #2 (Remarks to the Author):

Dear authors

The ms entitled "Can sub-Saharan Africa feed itself? The coming financial phosphorus bottleneck" brings important information on forecast for fertilizer demand to improve agriculture in SSA. This is an important message considering the auto-sufficient food production for next decades.

I have some concerns:

Lines 62-84 - it is not clear for me the definition of MVCR and AVCR. You mention many values for some countries, but maybe should be good to consider what are the ideal values for each country considered in your study.

Lines 258-259 - You said that PUE is 0.8-1.0 now in Africa. I don't know if you have considered this PUE in your calculations, but remember that if you increase the application rate this PUE will drop substantially, as we have seen over the last 2-3 decades in other countries like China, Brazil, etc...

best regards

Reviewer #3 (Remarks to the Author):

- What are the noteworthy results?

The manuscript describes the economics of phosphate fertilizer requirements for sub-Saharan Africa, showing the need for the period 2020-2050 and emphasising the difficulties in achieving that through existing economic practices.

The manuscript focuses on phosphate rock, and does not address the perhaps more widely traded phosphorus sources, such as DAP. It is also unusual to use phosphate rock on its own, as compound NPK fertilizers are used to get the balance right between N, P and K for specific soils and crops. So by focusing on phosphate rock, the approach taken (a) does not consider the use of phosphate rock to manufacture chemical fertilizers, via phosphoric acid, and (b) does not consider the use of artificial compound fertilizers, which are what is more likely to be used, from a practical point of view.

- Will the work be of significance to the field and related fields? How does it compare to the established literature? If the work is not original, please provide relevant references.

The work complements the existing literature on P fertilizers, but it does not take into account some significant work (e.g. Sheldrick and Lingard, Food Policy 29 (2004) 61-98) on nutrient balances in Africa. Nor does it develop the peak phosphorus debate, in which the reclassification of reserves by the USGS effectively extended P availability by decades; the concept of 'reserves' and 'resources' has not been properly understood by many in this debate.

- Does the work support the conclusions and claims, or is additional evidence needed?

The work as presented supports the conclusions that are made; all that is needed is presented, although an unfamiliar reader will have to read to the end and then re-read the manuscript to fully understand the evidence base that is used.

- Are there any flaws in the data analysis, interpretation and conclusions? - Do these prohibit publication or require revision?

I think there are some serious issues of definition that could be resolved more clearly. First, the use of the terms phosphate, P and phosphate rock is not rigorously defined from the start (although it comes in the final 'Terminology' section. This is potentially an issue in Figure 1, where the axis is labelled 'P (1000 tons)'. A metric unit, tonne, should be used instead of 'ton'. The caption states '1000 tons' and this is not needed if the axis is correctly labelled. But there is cause for confusion, as 1000 tonnes P is equivalent to how much phosphate rock? Assuming phosphate rock has 32% P₂O₅, which is equivalent to 14% P, approximately 7000 tonnes of phosphate rock are needed to supply 1000 tonnes of P. I'm not convinced this calculation has been done correctly. The equivalent values, or conversion factors between the different forms of describing P fertilizers, should be given, and a consistent terminology used throughout.

Figure 1(b) has an axis labelled 'vol (tons)' – what is this? Amount in tonnes? Tons is not a unit of volume.

- Is the methodology sound? Does the work meet the expected standards in your field?

The approach taken is sound, although the queries raised above need to be clarified.
The overall

- Is there enough detail provided in the methods for the work to be reproduced?

Yes, as long as the issue concerning consistency of units is clearly addressed.

David Manning

RESPONSE TO REVIEWERS' COMMENTS

Reviewer #1:

This manuscript quantifies the amount of phosphate rock required by 10 nations in Sub-Saharan Africa (SSA) to sustainably intensify agriculture by 2050 under business-as-usual scenario, and, these nations' relative expenditure on phosphate rock. It typifies these countries as following one of three economic phosphorus pathways, one of which is a 'financial phosphorus bottleneck', highlighting the scale of the challenges facing SSA.

Optimising sustainable phosphorus use in SSA is critical to the continent's food security and the viability of their food systems. The issues addressed by the manuscript are important, and much work is required in the field at both the macro and micro level to better understand the nature and implications of phosphorus-specific challenges in SSA for long-term food security. Few studies have addressed the macro issues, which this manuscript does. The manuscript is novel in the authors' linking of their macro analysis in this manuscript to their recently published geochemical dynamic phosphorus pool simulator model (GDPPS) (Magnone et al 2019). Some of the authors are certainly experts in the field of soil phosphorus, and in their previous 2019 paper they highlighted a) the important implications of soil chemistry on phosphorus use efficiency and local and regional scales, and b) that the heterogenous nature of SSA's soil chemistry has not been factored into key influential models used to predict phosphorus fertiliser use on the continent.

There are however some significant limitations of the manuscript in its' current form. These mainly pertain to fundamental (non-soil) assumptions around food security, phosphorus and policy, and the transparency of these, as outlined below.

Response: We thank the reviewer for the time they have spent assessing our work and the detailed feedback they have provided. The review is thorough and fair and we are grateful that they have clearly recognised the contribution we have made whilst also highlighting the limitations of the original draft. These limitations were mostly due to us overstepping our expertise (as the reviewer has highlighted) and bringing this to our attention it has helped us improve the work considerably.

In responses to this review, we have made the following changes (i) undertaken new modelling to provide new scenarios; (ii) added dietary discussion in relation to different scenarios; (iii) provided greater caveats on our key conclusions; and (iv) removed VCR discussion.

First, the study only models one single future scenario, rather than multiple. Why only use a business as usual scenario? Given the uncertainty and complexity surrounding future food security and land use, multiple scenarios would seem more appropriate.

R1.1 **Agreed, thank you, done.** The reviewer has highlighted an important point, and we have now conducted the modelling for SSP1 to 5. We have added the following section to highlight this and a new figure 1 (see figures below):

Line 184 (Introduction) - Therefore, the extent to which fertiliser, including phosphate, is used is controlled by socio-economic factors 32,33 the trajectory of which may take several different pathways. The conceptual framework of Shared Socioeconomic Pathways (SSPs) has been developed to account for human behaviour in the field of environmental and climatic modelling 34,35. It outlines five alternative plausible societal trends, which will have varying impacts on ecosystems and the economy: SSP1 represents sustainability and a "greener" future; SSP2 business as usual; SSP3 regional rivalries (i.e. increased nationalism); SSP4 is a divided an unequal society and SSP5 fossil fuelled development 36,37.

AND

Line 211 (Results and Discussion) - We projected Sub-Saharan Africa elemental phosphorus requirements using the IMAGE 3.0 database which, in turn, uses SSPs as an underlying framework^{34,35}. IMAGE 3.0 projected, crop production from, amongst other variables, population and dietary requirements, and we predicted the phosphorus requirement based on the amount of phosphorus within the given crops and the efficiency of the soil³⁸. Efficiency of soil was defined by the concentration of iron and aluminium oxides^{40,41}. We projected SSP1-5 at a continental level and assessed specific countries under SSP2, only.

Our results indicate that, regardless of SSP, Sub-Saharan Africa will require large increases in phosphorus application to maintain agricultural production in line with population growth other socioeconomic factors (Figure 1). In 2016, elemental phosphorus application via fertilisers was approximately 560,000 tonnes across the continent, however, by 2050, the lowest likely application of elemental phosphorus is 1.7 million tonnes, an increase of 310 %. This rate is under SSP4 which represents a more divided and unequal society and therefore is not a favourable scenario. The highest application rates are in SSP5 which represents a fossil fuelled economic growth model, here total continental phosphorus application is projected to be 3.4 million tonnes, a 620 % increase. The other SSPs (1,2 & 3) follow similar trajectories to each other ending with values of 2.6 million tonnes (475 %), 3.1 million tonnes (550 %) and 3.0 million tonnes (530 %) respectively.

Second, this business as usual scenario is based on the Shared Socioeconomic Pathways (SSP2). Why use SSPs rather than other future scenarios? The SSPs were designed to inform the analysis of climate impacts, vulnerability mitigation and adaptation. Whilst they do include land-use, achieving food security and sustainable food systems goes far beyond land-use issues. There is therefore a key missing link in this manuscript to the key current food security narrative. The global food security discourse is shifting beyond ‘sustainable intensification’ to a critical focus also on ‘healthy diets’ and ‘sustainable food systems’. For example, the EAT-Lancet Commission of 37 organisations developed the “planetary health diet” (published in *The Lancet*). Transitioning to healthy diets within planetary boundary limits by 2050 will require substantial dietary shifts. Business as usual agricultural production systems will no longer be an option for the future. E.g. “Strategy two: reorient agricultural priorities from producing large quantities of food to producing healthy food”. This means there is a need to transform food systems and grow what we need for healthy diets, rather than just making the current fundamentally flawed food system more ‘efficient’ through intensification. Sustainable intensification is important, but it alone cannot lead to food security and sustainable food systems. E.g. in SSA, a significant increase in consumption of vegetable, fruits, whole grains, nuts, and dramatic decrease in starchy vegetables, needs to take place for healthy diets, rather than improving the efficiency of the current crops under production. What would be the phosphorus demand in SSA countries under these production systems? EAT-Lancet has indeed answered part of this question, though the authors of this Nature manuscript could improve it with their soil model (in addition to the EAT-Lancet Commission refs - Willett et al and Springmann et al, see also the World Resources Institute and CGIAR’s International Food Policy Research Institute, references below – Ranganathan et al and Robinson et al)

R1.2 Agreed, thank you, done. We did not acknowledge the importance of diets within this manuscript and focussed too heavily on sustainable intensification. To address we have amended the manuscript away from the focus on *food security* and focussed instead on *agricultural production* (as suggested by the reviewer later). We have added a paragraph in the introduction on diets using the references kindly provided (and others) and we have added a discussion on how the different SSPs (added at R1.1) relate to this. These are covered:

Line 50 (Introduction) – The 21st century faces a “food gap”: the estimated 70 % difference between the crop calories available in 2006 and expected calorie demand in 2050. One of the key agricultural production challenges is, paradoxically, the projected increased wealth for most citizens¹. This is because such increases in wealth are associated with higher calory intake, and often, with increased consumption of resource intensive foods in diets (e.g. meat and dairy)². Historically, increases in demands have been met by increased agricultural production, but this method of production comes at a high ecological and

environmental cost¹. Today the challenge is how to meet calory demand whilst minimising ecological and environmental damage.

This challenge is perhaps greatest in Sub-Saharan Africa, where 23 % of the population suffers from chronic hunger^{1,3}. In this region, calory intake is set to increase dramatically between 2020 and 2050: in Nigeria intake is expected to increase from 2700 to 3000 kcal capita⁻¹ day⁻¹ (110 %) and in Ethiopia from 2100 to 2600 kcal capita⁻¹ day⁻¹ (123 %) and all amongst a growing population. By contrast, the US and Europe already have a mean calory intake of 3500 kcal capita⁻¹ day⁻¹ which is predicted to maintain constant for the duration. This means that both quality and quantity of increased food production are critical developing a food secure future.¹

AND

Line 106 (Introduction) – Globally, it is estimated that the phosphorus footprint of 2050 will be 1.5 times that of 2010 due to population increase and dietary change^{19,20}

AND

Line 182 (Introduction) – Therefore, the extent to which fertiliser, including phosphate, is used is controlled by socio-economic factors^{32,33} the trajectory of which may take several different pathways. The conceptual framework of Shared Socioeconomic Pathways (SSPs) has been developed to account for human behaviour in the field of environmental and climatic modelling^{34,35}. It outlines five alternative plausible societal trends, which will have varying impacts on ecosystems and the economy: SSP1 represents sustainability and a “greener” future; SSP2 business as usual; SSP3 regional rivalries (i.e. increased nationalism); SSP4 is a divided an unequal society and SSP5 fossil fuelled development^{36,37}.

These scenarios have been built into the Integrated Model to Assess the Global Environment (IMAGE 3.0) assessing land use, including agricultural land use, change on a gridded scale^{38,39}. IMAGE3.0 has projected crop production, agricultural efficiency, land use change, food prices and trade for the 21st Century within the SSP framework. Such projects are based upon human factors such as population, GDP, policies, technology, lifestyle (e.g. dietary change) and resources and physical factors such as accessibility, terrain and soil properties. IMAGE 3.0 indicates that, regardless of scenario, Sub-Saharan Africa will require the largest expansion in agricultural land on the planet and, in SSP3 (i.e. increased nationalism) and 4 (i.e. a divided an unequal society), increasing food import dependency. This is driven by changes in food consumption which vary depending on per capita GDP and agricultural efficiency. The sustainable pathway, SSP1, is achieved through increasing agricultural efficiency which reduces the requirement to expand agricultural lands³⁸.

At least some (perhaps all) of this manuscript’s authors are very much experts in their field of soil modelling and phosphorus, and it would be a tremendous contribution to new knowledge to model more relevant sustainable food scenarios. The EAT-Lancet Commission’s work could be improved/built-upon to show what are the implications of shifting healthy diets (in a low food waste etc scenario) on phosphorus use and expenditure in SSA? To my knowledge, EAT-Lancet Commission did model the phosphorus implications of both BAU and healthy diets in 2050, but perhaps did not include the soil phosphorus complexities that the Nature manuscript authors highlight. It may be that phosphorus demand needs to increase even more than using the SSP2 scenario. This would be valuable new knowledge with important implications. It would be interesting (and arguably necessary) to compare the manuscript authors BAU scenarios for 2050 with the EAT-Lancet projections for 2050.

R1.3 Agreed, thank you, done. In response to R1.1 by including all five SSPs we have expanded this BAU. We recognise that we did not clearly explain the dietary etc. assumptions behind the IMAGE 3.0 dataset we have rectified this by adding the following text.

Line 191 (Introduction) – These scenarios have been built into the Integrated Model to Assess the Global Environment (IMAGE 3.0) assessing land use, including agricultural land use, change on a gridded scale

^{38,39}. IMAGE3.0 has projected crop production, agricultural efficiency, land use change, food prices and trade for the 21st Century within the SSP framework. Such projects are based upon human factors such as population, GDP, policies, technology, lifestyle (e.g. dietary change) and resources and physical factors such as accessibility, terrain and soil properties. IMAGE 3.0 indicates that, regardless of scenario, Sub-Saharan Africa will require the largest expansion in agricultural land on the planet and, in SSP3 (i.e. increased nationalism) and 4 (i.e. a divided and unequal society), increasing food import dependency. This is driven by changes in food consumption which vary depending on per capita GDP and agricultural efficiency. The sustainable pathway, SSP1, is achieved through increasing agricultural efficiency which reduces the requirement to expand agricultural lands ³⁸.

AND

Line 228 (Results and Discussion) These results, most likely, reflect the socioeconomic changes within Sub-Saharan Africa. SSP4 “the unequal society”, along with SSP3 “increased nationalism”, have the highest population growth trajectories ³⁸, yet SSP4 has the lowest phosphorus requirements (Figure 1). This is because SSP4 has low GDP growth – due to the high levels of inequality ³⁸ – meaning that large proportions of the population are unlikely to adopt the expensive and resource rich “western diet” ^{1,3}. By contrast SSP1 “the sustainable pathway” and SSP5 “extractive industrial growth” have the lowest population growth ³⁸, yet SSP5 has the highest phosphorus requirements (Figure 1). Again, this is likely to be because growth in GDP is rapid under these scenarios – as is resource use more generally – and thus a large proportion of the population is of likely to adopt the resource rich “western diet” ³.

The high phosphorus consumption within SSP1 is particularly interesting, since this is considered to be the “sustainable” pathway ³⁷. However, when viewed in terms of economic growth this scenario is the most sustainable. SSP1 has the second highest increase in GDP of all scenarios (after SSP5) ³⁸, yet by 2050, this pathway will use 24 % less phosphorus than the SSP5 and 16 % less than SSP2 “business as usual”, and SSP3 “increased nationalism” (Figure 1). Only SSP4 “the unequal society” uses less phosphorus than SSP1 because under SSP4 the majority of lifestyles to not improve ^{1,3}. Therefore, SSP1 is the most phosphorus efficient pathway to deliver economic growth and improvements in lifestyle.

A third important assumption behind this manuscript which needs to be clarified, and possibly revised, relates to phosphate rock concentration. The authors assume the concentration of phosphate rock is constant to 2050. However we know that as a finite resource, the average concentration of phosphate rock is and will continue to decline over the long-term as the high-grade phosphate rock is mined out (e.g. IFDC 2010). This should be included in the model, as it can significantly affect quantities and cost. The authors also use a surprisingly high average phosphate rock concentration of 35% P₂O₅ (350g of P₂O₅ per kg of rock), noted in lines 299-301 (p11). There are few existing deposits with this high grade, and certainly in the future this will decline. [further, this looks like a slight calculation error, as P₂O₅ = 0.4576 BPL, so the author’s assumption of 70% BPL is actually equal to 32% P₂O₅, not 35% P₂O₅. This may also distort the results a little].

R1.3 Clarified, thank you. This was a typographical error rather than calculation error on our part. on our part all calculations were done at 32% P₂O₅. This has been corrected.

Fourth, and related to phosphate concentration, it appears the authors have chosen a static price for phosphate rock to 2050. Phosphate rock mining and processing costs will almost certainly increase over the long term, as new mine investments must be made, processing lower-grade phosphate rock in harder to reach locations and containing higher impurities (IFDC 2010). There are many uncertainties and debates around the longevity of phosphate rock, but the decline in concentration and increase in costs is not disputed. The cost per tonne of elemental phosphorus will increase, hence price, and this should therefore be reflected in the analysis presented here, rather than assuming costs will be stable through to 2050. It will be interesting to see how sensitive the authors’ analysis of economic pathways are to a long-term price increase in phosphate.

R1.5 Agreed, thank you, done. We agree we should have reflected reasonable ranges within our calculations. We have taken a value of mean value of US\$ 115 ± 33 this is derived from the mean price for phosphate rock between 2010 and 2020. We have propagated the errors to provided likely ranges for phosphate cost and expenditure. We have added a new figure (Fig. 3) to show this and the following text:

We have provided additional information on phosphate rock price history in the introduction:

Line 126 (Introduction) It is in developing countries where fertiliser use is increasing most rapidly ²⁴, therefore, both the amount of phosphate rock required and the cost of this is important to understanding phosphate processes in countries where agricultural intensification is required. Phosphate rock price, as a limiting factor, became most acute during the phosphate crisis of 2008. Between, 1960 to 2005 phosphate rock had a relatively stable but increasing commodity price of US\$ 31 ± 36 per tonne (including a minor spike in 1975) ²⁵. But in 2008, prices increased by 800 % from US\$ 50 to US\$ 50 430 due to a range of short-term issues resulting in demand outstripping supply ^{13,16}. By the end of 2009 prices had reduced, though not to pre-spike limits, and the price during the 2010 to 2020 decade has been US\$ 115 ± 33 per tonne of phosphate rock – mostly decreasing. This means that the last decade has had both the highest sustained phosphate rock price and greatest fluctuations of any decade since 1960 ²⁵.

Over the coming decades, there are several factors which might put downward pressure on the price, for example, even conservative estimates expect supply of phosphate rock to increase in the coming decades ²⁶ and, technological developments like recycling elemental phosphorus – “closing the phosphorus loop” – and more efficient rock processing (i.e. lack of wastage) will reduce demand for phosphate rock. On the other hand, as production moves to lower grade material – with high levels of contaminant metals – processing costs will increase. The most unpredictable element of phosphate cost is geopolitical tensions ²⁷.

AND

Line 308 (Results and Discussion) - We project the total cost to Sub-Saharan Africa between 2020 and 2050 to be US\$ 50 ± 15 billion. For the four highest consuming countries, Nigeria, Ethiopia, South Africa and Ghana the 2050 expenditure is projected to be 504 ± 147, 325 ± 95, 101 ± 29 and 146 ± 43 million US\$ per year, respectively, up from 53 ± 43, 74 ± 32, 70 ± 21 and 11 ± 15 million US\$ per year, respectively, between 2010 and 2020. However, these figures do have important caveats. The first caveat is that, our projections assume that phosphate rock price stays within the limits of the price during the 2010 to 2020 decade (US\$ 115 ± 33 per tonne of phosphate rock) – which represents the decade of highest sustained and most fluctuating prices. The second caveat is that, this is not a calculation of the farmgate price, which is usually considerably higher than the phosphate rock price due to additional costs discussed in the introduction (e.g. processing, taxes, transportation etc.). In some areas the price will be double our estimate due to the addition – non-production – costs ³¹. Nevertheless, focussing on phosphate rock price provides a constant “floor” price under which relative phosphate expenditure will not fall, and whilst not as directly applicable as farmgate prices provides context and may be viewed as a *better than best case* scenario.

We assessed phosphate rock expenditure as a percentage of independently projected real GDP for each country ⁴⁴. Real GDP adjusts for inflation of goods and services by an economy for a given year and is expressed at base year prices. These results indicate that the studied nations have very different phosphate rock cost pathways. Between 1990 and 2020, Nigeria, South Africa and Ghana all had extremely low phosphate rock cost to GDP ratios (<0.05 %), which, in the case of South Africa, was part of a steady descent from a high of 0.2 in the late 1970s (Figure 3D). Nigeria and Ghana, on the other hand have not had a high phosphate cost to GDP ratio and the coming decades are predicted to be the highest they have experienced. These are projected to peak in 2030 with values of 0.05 ± 0.01 % and 0.12 ± 0.03 %, respectively. Ethiopia has experienced the highest historic phosphate rock cost to GDP ratio of any of the studied nations with a peak of 0.35 % during the phosphate crisis of 2008 (Figure 3D) which also saw an increase in phosphate use in the country (Figure 3B). Over the 2010 to 2020 decade there has been a

decrease in the ratio to approximately 0.1 % and the projected future values of 0.12 ± 0.04 % and 0.09 ± 0.03 % in 2030 and 2050, respectively, are likely to represent a continuing of this plateauing.

The results from South Africa, and to a lesser extent Ethiopia, indicate a potential pattern of phosphate expenditure relative to GDP. The first part of this pattern is a low phosphate expenditure relative to GDP. The second part sees the use of phosphate increase (Figure 3B) which drives up relative expenditure (Figure 3C) reaching a peak in the ratio (Figure 3D). Evidence from South Africa and Ethiopia suggests that time spent at the peak is relatively short and precedes a third phase where expenditure declines to a plateau (Figure 3D). The decline in this ratio is driven by an increase in the GDP⁴⁵ and, in the case of South Africa, the plateau is lower than at the start of this process (Figure 3D). Overprinted onto this process are global trends, the most important of which is the phosphate price, as evidenced by the extremely high peak in Ethiopia in 2008 (Figure 3D). We predict that Nigeria and Ghana will experience this peak around the year 2030 with Ghana experiencing a considerably higher peak than Nigeria (Figure 3D).

Fifth, still on the topic of costs, the study looks at SSA nations expenditure of phosphate rock – but what about expenditure of phosphate fertilisers? (including fertiliser blends that include P among other nutrients?). Some African countries import finished fertiliser products (and re-blend to suit local conditions). Phosphate fertiliser costs are obviously also critical to this study. How have these been accounted for? What are the different phosphorus production and import profiles of the 10 SSA nations studied? Is most phosphorus demand in these countries met by imported mixed fertilisers, supplemented with a small amount of domestic phosphate rock production, and a small amount of imported phosphate rock imports? South Africa may be an anomaly as a large phosphate rock producing country. The point is, how do these different sources of phosphorus in these SSA countries affect the price and hence the results of the analysis?

R1.6 Clarified, thank you. We believe there is some confusion arising from a lack of clarity in our writing. When we refer to phosphate rock we are referring to phosphate rock as a raw material which can be other phosphate fertiliser products (e.g. NPKs, DAP, MAP and TSPs): we are not referring to the small amount of phosphate rock which is applied directly to soils. We have clarified this in considerable detail in response to reviewer 3 (R3.1) who requested detail on the same topic. We believe our response to your next point (R1.7) clarifies the limitations of costs which are highlighted here.

Additionally, we agree it was an oversight not to include phosphate rock reserve values and importation levels. We added the following text to address to describe this:

Line 291 (Results and Discussion) – Whether this phosphate rock will be produced domestically or is imported is beyond the scope of this study. However, for context, since 2010 Nigeria, Coté d'Ivoire⁴², Ghana^{42,43} and Ethiopia⁴³ have all imported most of the phosphate rock used. By contrast, in 2010 South Africa produced 2.3 million tonnes of phosphate rock domestically – a net surplus – and Mali imported only negligible amounts of phosphate rock¹⁵. By 2050 under SSP2, the current South African extraction rate would still provide a domestic surplus, albeit much diminished. Furthermore, whilst estimates of the South African phosphate reserve range from 230¹⁵ to 1,400 million tonnes²³ both far exceed the total phosphate rock required: 42 million tonnes over the period to 2050 (Figure 3A). This indicates that whilst South Africa will require substantial increases in application this is within the limits of the domestic phosphate reserve, by contrast Nigeria, Ethiopia, Ghana and Coté d'Ivoire are much more likely to rely in importation.

However, our results do demonstrate that the total projected Sub-Saharan African phosphate rock usage between 2020 and 2050 (440 million tonnes, Figure 3A) represents less than 1 % of current known global reserves (65 billion tonnes²³). Therefore, it will be economic factors rather than physical factors that limit availability of this resources.

Sixth, continuing on cost assumptions, how have the authors accounted for 'farm gate' cost of phosphate fertilisers? We know that farm-gate prices in land-locked African countries can be substantially higher than

other parts of the world (e.g. 2-5 times higher due to transport, port duties, taxes, etc, see IFDC 2007). If the authors are using VCR to estimate affordability and “the extent to which fertiliser, including phosphate, is actually used on farmers” (line 60), then farm gate prices would be important to use. The costs in Figure 1 don’t appear to consider farm gate prices? Further, farm gate prices would vary from country to country, so Figure 1B graph would not be able to use the same bars to match the different Y Axes of tons and cost.

R1.7 Agreed, thank you, clarified. We recognise that the price farmers pay at the farm gate is affected by prices other than the phosphate rock market price. However, we were unable to compile an accurate reflection of the farm gate price in all our key countries. To give two examples of the key difficulties: (1) the IFDC (2007), report which the reviewer highlights, focusses on 6 countries only two of which are in our key 10 countries; (2) Bonilla Cedrez et al., (2020, PLOS One) provided analysis on more countries but focussed on urea rather than phosphate rock. In both, cases we felt this did not provide sufficient coverage. Therefore, we chose the phosphate market price because of the universality of the price.

However, we agree it is important to highlight this limitation and have done so by adding the following text in the discussion.

Line 151 (Introduction) - Furthermore, farm-gate phosphate fertiliser prices vary considerably across the continent due to non-production costs. For example, in 2006, production costs of DAP made up just 51 – 53 % of the farmgate price in Mali and 59 % in Uganda but between 66 and 75 % in Tanzania. Transport costs were the next highest contributor to costs and this was highest in inland areas – particularly those with poor infrastructure ³¹.

AND

Line 314 (Results and Discussion) – The second caveat is that, this is not a calculation of the farmgate price, which is usually considerably higher than the phosphate rock price due to additional costs discussed in the introduction (e.g. processing, taxes, transportation etc.). In some areas the price will be double our estimate due to the addition – non-production – costs ³¹. Nevertheless, focussing on phosphate rock price provides a constant “floor” price under which relative phosphate expenditure will not fall, and whilst not as directly applicable as farmgate prices provides context and may be viewed as a *better than best case* scenario.

Finally, the authors briefly touch on policy implications in lines 149-159. In the manuscript’s current form, these important implications are not sufficiently explained/discussed, and they read as rather simplistic and somewhat inappropriate. E.g. the authors suggest that the only way forward for countries on the ‘challenging’ pathway are one of 2 options: either improve MVCR (e.g. through subsidies or transport improvements), “or else, the countries will have to supplement low yields with high rates of imported food” (lines 157-158). There is a whole myriad of other possible policy options that are not discussed here, possibly because the modelled assumptions themselves are so narrow, or due to the authors areas of expertise lying elsewhere. These policy implications should therefore be attended to in more expert and nuanced detail, or perhaps removed.

R1.8 Agreed, thank you, rectified. Again, this was extrapolation beyond our expertise. To rectify this, we have removed the policy implications. The reviewer is correct, this was crude and did not enhance the study.

The authors are clearly experts in agricultural soil nutrient dynamics, as indicated by the level of attention given to these aspects (such as the entire Supplementary Information). However there are also many complexities, nuances and uncertainties in the fields of food security, phosphorus economics and policy implications, some of which have been highlighted above and many of which have not been addressed appropriately in this manuscript. The cost assumptions appear to over-simplify a very complex and dynamic situation, for example. The study would therefore at minimum need to demonstrate the sensitivity of the analysis to the variations and uncertainties indicated above, and address the specific comments below. The

analysis would benefit from using multiple scenarios, and, scenarios that go beyond sustainable intensification to address the current food systems discussions around what would it take to achieve healthy diets and hence food security. The study needs to link to related research, such as the EAT-Lancet Commissions work, which does model phosphorus scenarios to 2050 (see also their Supplementary Material linked below). If the authors do not want to get into the food security debate, then they should significantly modify and qualify the stated intentions and findings of this study to refer to improving agricultural productivity, NOT food security.

R1.9 Agreed, thank you, done. We did conflate “food security” with “agricultural production”. We have corrected the terminology throughout. As highlighted in R1.3 we have added reference to EAT-Lancet Commission’s work (with reference to the papers the reviewer has highlighted). The changes made at R1.8 also address this.

References:

Prof Walter Willett, MD, Prof Johan Rockström, PhD, Brent Loken, PhD , Marco Springmann, PhD, Prof Tim Lang, PhD, Sonja Vermeulen, PhD, et al. (2019), ‘Food in the Anthropocene: the EAT–Lancet Commission on healthy diets from sustainable food systems’ THE LANCET COMMISSIONS| VOLUME 393, ISSUE 10170, P447-492, FEBRUARY 02, 2019 <https://www.thelancet.com/commissions/EAT> Willett et al Supplementary material: [https://www.thelancet.com/journals/lancet/article/PIIS0140-6736\(18\)31788-4/fulltext#supplementaryMaterial](https://www.thelancet.com/journals/lancet/article/PIIS0140-6736(18)31788-4/fulltext#supplementaryMaterial)

Janet Ranganathan, Daniel Vennard, Richard Waite, Brian Lipinski et a; (2016), ‘Shifting Diets for a Sustainable Food Future’, Creating a Sustainable Food Future, Installment Eleven. World Resources Institute https://files.wri.org/s3fs-public/Shifting_Diets_for_a_Sustainable_Food_Future_1.pdf

IFDC (2007). ‘Fertilizer supply and costs in Africa’, prepared by Chemonics International Inc. and the International Center for Soil Fertility and Agricultural Development.

IFDC (2010). ‘World phosphate rock reserves and resources’. Tech. Bull. T-75. International Fertilizer Development Centre (IFDC). Washington.

SHERMAN ROBINSON, DANIEL MASON D’CROZ, SHAHNILA ISLAM, TIMOTHY B. SULSER, RICHARD D. ROBERTSON, TINGJU ZHU, ARTHUR GUENEAU, GAUTHIER PITOIS, MARK W. ROSEGRANT (2015), The International Model for Policy Analysis of Agricultural Commodities and Trade (IMPACT): Model description for version 3, IFPRI DISCUSSION PAPER. 2015 <https://www.ifpri.org/publication/international-model-policy-analysis-agricultural-commodities-and-trade-impact-model-0>

Springmann, Marco; Clark, Michael; Mason-D’Croz, Daniel; Wiebe, Keith; Bodirsky, Benjamin Leon; Lassaletta, Luis; de Vries, Wim (2018) ‘Options for keeping the food system within environmental limits. (Technical report)’ Nature, 2018, Vol.562(7728), p.519

Specific comments:

Line 27-28: There are a few value-laden bold statements made in the manuscript, which need to be tempered and qualified, given the complexities and uncertainties highlighted above. E.g. “SSA requires a 2.5-fold increase in food production during the 21st century, in order to become food secure”. This prediction is very much under the ‘current’ business as usual production paradigm. Even if production and productivity increase, there is no guarantee SSA food consumers have financial and physical access to healthy diverse foods essential for food security (not to mention the current production paradigm has ~40% food losses high up in the value chain). We know business as usual is not an option and food systems need to dramatically transform. This has serious implications for phosphorus demand, as discussed in general comments.

R1.9 **Agreed, thank you, done.** We agree this sentence was expressed too assertively. We have changed that this sentence to reflect both that it is a BAU case and an estimate from a single study. We also have added two new paragraphs at the start of the manuscript to give greater context (this is provided at R.1 and Line 50).

Line 70-76: Many of these VCRs are from different sources/references. How comparable are they? Is there likely to be wide variation between them due to the reference year? Or due to different assumptions and data sources/reliability? You have noted in the paragraph below that VCRs can change dramatically year to year.

R1.10 **Agreed, thank you, done.** In response to R1.8 we removed reference to VCR.

Line 73-76: This list in brackets reads a little awkward in the text. Perhaps put in a simple table for ease of readability, including year of VCR for transparency.

R1.11 **Agreed, thank you, done.** In response to R1.8 we removed reference to VCR.

Line 115: Given these are future projections and scenarios, best to say “Our results suggest that...”, or “These results indicate that ...are likely to have very different phosphorus pathways”.

R1.12 **Agreed, thank you, done.** We have changed “show” to “indicate”.

Line 122-125 (FIGURE 1): Figure 1 appears to be in a very draft form with quite a few errors:

> First, there are 3 graphs (A, B, C) but only A and B are mentioned in the figure heading. In the Figure heading, (A) should refer to (B), and (B) should refer to (C). The first sentence in line 123 should refer to (A).

> Second, Graph A shows 2020-2050 while Graphs B and C show 2018, 2030, 2050. Obviously it would be better if these date ranges were consistent.

> Third, Graphs B and C should indicate whether 2018 data is actual or modelled, to get a sense of comparability with 2030 and 2050.

R1.13 **Agreed, thank you, done.** In response to the three comments above, we have drawn this this figure completely (please see end of this letter for new figures). We have added the line “as reported by the World Bank For 2018 and predicted for 2030 and 2050” to clarify this. We have also corrected this in Table 1 in the methods.

> Fourth, In Graph A, 2 of the bars exceed the Y axis scale, and in Graph C, 2 of the bars in 2030 exceed the Y Axis scale. The scale should obviously be extended.

R1.14 **Agreed, thank you, done.** We have corrected this.

Line 300-301: “We assume it to be 70% BPL which is approximately equivalent to 350g of P₂O₅ per kg of rock”. This is slightly incorrect. I assume the authors meant 320g of P₂O₅. They have noted the correct calculation on line 235 “70% BPL ~ 32% P₂O₅”.

R1.15 **Agreed, thank you, done.** The reviewer is correct, this was a typographical error. The at line 300, now corrected, the calculations were undertaken using the correct value.

Reviewer #2:

Dear authors

The ms entitled "Can sub-Saharan Africa feed itself? The coming financial phosphorus bottleneck" brings important information on forecast for fertilizer demand to improve agriculture in SSA. This is an important message considering the auto-sufficient food production for next decades.

Response: Thank you for taking the time to review our work and recognising the importance of what we have produced.

I have some concerns:

Lines 62-84 - it is not clear for me the definition of MVCR and AVCR. You mention many values for some countries, but maybe should be good to consider what are the ideal values for each country considered in your study.

R2.1 Agreed, thank you, done. We have removed reference to MVCR and AVCR because (a) the lack of coverage and variations highlighted by the reviewer; (b) the focus of this paper was supposed to be on phosphorus requirements and only touch on the costs but the reviews have shown we focussed too heavily on the economics.

Lines 258-259 - You said that PUE is 0.8-1.0 now in Africa. I don't know if you have considered this PUE in your calculations, but remember that if you increase the application rate this PUE will drop substantially, as we have seen over the last 2-3 decades in other countries like China, Brazil, etc...

R2.2 Clarified, thank you. This value was provided for information only and is not included in the modelling. This was not clear, we have added "these do not have an effect on our model" (line 480) to clarify.

best regards

Reviewer #3:

- What are the noteworthy results?

The manuscript describes the economics of phosphate fertilizer requirements for sub-Saharan Africa, showing the need for the period 2020-2050 and emphasising the difficulties in achieving that through existing economic practices.

Response: Thank you for taking the time to review our work. We are grateful for the detail that you have provided which in implementing has greatly improved our manuscript.

The manuscript focuses on phosphate rock, and does not address the perhaps more widely traded phosphorus sources, such as DAP. It is also unusual to use phosphate rock on its own, as compound NPK fertilizers are used to get the balance right between N, P and K for specific soils and crops. So by focusing on phosphate rock, the approach taken (a) does not consider the use of phosphate rock to manufacture chemical fertilizers, via phosphoric acid, and (b) does not consider the use of artificial compound fertilizers, which are what is more likely to be used, from a practical point of view.

R3.1 Clarified, thank you. We believe there is some confusion arising from a lack of clarity in our writing. When we refer to phosphate rock we are referring to phosphate rock as a raw material which can be other phosphate fertiliser products (e.g. NPKs, DAP, MAP and TSPs): we are not referring to the small amount of phosphate rock which is applied directly to soils. We have clarified this below and with additional points about costs in response to reviewer

Line 83 (Introduction) – The UN Food and Agricultural Organisation (FAO) recognises 13 essential mineral macro- and micronutrients for plant growth (with nickel excluded due to debate as to whether it is essential or beneficial). Of these, nitrogen, phosphorus, and potassium are considered the most yield limiting¹⁰.

By contrast to the more economically advanced regions, African countries have had low nutrient applications throughout the 20th Century and instead of building up residual soil nutrient reserves, Sub-Saharan Africa countries have been mining the soils for decades¹¹. At the continental scale, nitrogen, phosphate and potassium have been severely depleted in soils between 1961 and 1998 with depletion rates for increasing by 225 %, 233 % and 256 %, respectively¹². The low crop yields in Sub-Saharan Africa have partially been attributed to this⁷.

By weight, phosphorus generally contributes less than 10 % to the total nutrient uptake to most African crops contrasting with nitrogen which is often about 50 %¹². Nevertheless, the FAO describes phosphorus as the nutrient with the greatest potential to limit agricultural yield enhancement in the future¹⁰. This is because, phosphorus is a critical nutrient which limits plant growth and there are no known biochemical alternatives – hence its description as “life’s bottleneck”^{13,14}. There are an array of phosphate fertiliser products, the main being NPKs, DAP, MAP and TSPs, however the primary source of phosphate within all of these is phosphate rock¹⁵ which is a finite resource^{16,17}. Globally, about 20 % of mined phosphate rock is lost during mining and distribution processes prior to its application to arable soils¹⁸. Our reference to phosphate rock, within this paper, is as raw material which can be manufactured to fertiliser and not, the small proportion which is applied directly to soils.

Globally, it is estimated that the phosphorus footprint of 2050 will be 1.5 times that of 2010 due to population increase and dietary change^{19,20}. As such, there has been considerable debate on the rate of phosphate rock depletion, often centring on the question of “*will we run out?*” This has been discussed vigorously using the concept of “*peak phosphorus*”^{16,17} – analogous to Hubbert’s concept of “*peak oil*”²¹. However, strictly this debate is not about phosphorus (which is never lost from the Earth’s chemical cycles and, hence, cannot be depleted) but phosphate rock – which is the finite resource and which can be depleted¹⁶. To emphasise this distinction, within this article we refer to elemental phosphorus, phosphate, and phosphate rock separately.

The concept of peak phosphorus originated in the late 2000s. In 2009, Cordell *et al.*²² published an influential estimate that predicted peak production would occur in 2030 and that there were between 50 and 100 years of phosphate rock supplies left. However, a year later, in 2010, a major reassessment increased the estimate of global reserves from 15 billion tonnes to 65 billion tonnes by reevaluating previously overlooked decades old reports¹⁵. On this estimate, which remains unchanged and is acknowledged by the US Geological Survey²³, peak phosphorus would occur around 2070 - 2080¹³. Nevertheless, despite the increased estimate, there are still challenges, including that 85 % of the reserves are in a disputed territory between Morocco and Western Sahara making the extremely geopolitically vulnerable (both due to national monopolies and political instabilities). Furthermore, uncertainty remains about the grade of reserves remaining: low and hard to access grades will limit accessibility to the resource^{14,17}.

AND

Line 301 (Results and Discussion) – These results do, however, assume that the system of phosphate rock mining to phosphorus application onto arable soils is 100 % efficient. Currently, this part of the system is 80 % efficient¹⁸ therefore our results may underestimate by approximately 20 %, however, we do not know how efficiency will change over the next three decades. Nevertheless, our results do demonstrate that the total projected Sub-Saharan African phosphate rock usage between 2020 and 2050 (440 million tonnes, Figure 3A) represents less than 1 % of current known global reserves (65 billion tonnes²²). Therefore, it will be economic factors rather than physical factors that limit availability of this resources.

- Will the work be of significance to the field and related fields? How does it compare to the established literature? If the work is not original, please provide relevant references.

The work complements the existing literature on P fertilizers, but it does not take into account some significant work (e.g. Sheldrick and Lingard, Food Policy 29 (2004) 61-98) on nutrient balances in Africa. Nor does it develop the peak phosphorus debate, in which the reclassification of reserves by the USGS effectively extended P availability by decades; the concept of ‘reserves’ and ‘resources’ has not been properly understood by many in this debate.

R3.2 Agreed. thank you, done. We agree, that we should have spent more time discussing the peak phosphorus debate and dismissed this too quickly. We have address nutrient balances in our response to R3.1

- Does the work support the conclusions and claims, or is additional evidence needed?

The work as presented supports the conclusions that are made; all that is needed is presented, although an unfamiliar reader will have to read to the end and then re-read the manuscript to fully understand the evidence base that is used.

R3.3 Agreed. thank you, done. The changes made in response to all reviews have improved our manuscript.

- Are there any flaws in the data analysis, interpretation and conclusions? - Do these prohibit publication or require revision?

I think there are some serious issues of definition that could be resolved more clearly. First, the use of the terms phosphate, P and phosphate rock is not rigorously defined from the start (although it comes in the final ‘Terminology’ section. This is potentially an issue in Figure 1, where the axis is labelled ‘P (1000 tons)’. A metric unit, tonne, should be used instead of ‘ton’. The caption states ‘1000 tons’ and this is not needed if the axis is correctly labelled. But there is cause for confusion, as 1000 tonnes P is equivalent to how much phosphate rock? Assuming phosphate rock has 32% P₂O₅, which is equivalent to 14% P, approximately 7000 tonnes of phosphate rock are needed to supply 1000 tonnes of P. I’m not convinced this calculation has been done correctly. The equivalent values, or conversion factors between the different forms of describing P fertilizers, should be given, and a consistent terminology used throughout.

R3.4 **Agreed. thank you, done.** In response to R.3.2 we have improved clarity by (i) referring to either “elemental phosphorus”, “Phosphate Rock” or “Phosphate Fertiliser” consistently; (ii) we have changed to metric “tonnes” on the figures.

Clarified. thank you. We have checked our calculations, and these are correct and to demonstrate this we will use the example of South Africa. We project that South Africa will need 4300 tonnes of elemental phosphorus between 2020 and 2050. This requires 30.7 million tonnes of phosphate rock (at 32% P₂O₅ equivalent to 14 % elemental phosphorus – as the reviewer states). If this evenly distributed across each of the 30 years between 2020 and 2050 (which is broadly the case in South Africa) then this is approximately 1 million tonnes per year as shown in figure 3b. Figure 3 replaces the older Figure 1 and has been repositioned in the manuscript.

Figure 1(b) has an axis labelled ‘vol (tons)’ – what is this? Amount in tonnes? Tons is not a unit of volume.

R3.5 **Agreed. thank you, done.** We have produced a new figure (now Figure 3) and corrected “tons” to “tonne” and “vol” to “total” in this.

- Is the methodology sound? Does the work meet the expected standards in your field?

The approach taken is sound, although the queries raised above need to be clarified.

R3.6 **Agreed. thank you, done.** These have been clarified.

The overall

- Is there enough detail provided in the methods for the work to be reproduced?

Yes, as long as the issue concerning consistency of units is clearly addressed.

R3.6 **Agreed. thank you, done.** Consistency is addressed in R3.4

David Manning

Additional Changes

Due to the large changes made in this response to these reviews have added a new Abstract and Conclusion

Line 9 (Abstract) - Between 2020 and 2050 Sub-Saharan Africa must increase food production more than any other continent, historically, globally, this has been achieved this through intensification methods including the increased use of fertilisers. Phosphate fertilisers are a key component of this and yet are a finite resource – formed from mined phosphate rock. Furthermore, their cost has prevented the widespread use in Sub-Saharan Africa. Here, we project phosphate requirements for Sub-Saharan African within the framework of five shared socioeconomic pathways (SSPs). We demonstrate that Sub-Saharan Africa will require large increases in phosphate fertiliser application regardless of future trajectory with our estimates ranging from 310 % to 620 %. The lowest of these increases (310 %) is only achieved under an undesirable pathway (SSP4) where inequality increases and few people develop the wealth-levels required to sustain an expensive, resource intensive “western” diet. The highest increase in phosphorus use, 620 %, is where high levels of economic growth are stimulated by the extractive industries (SSP5). The other three pathways tested: sustainable (SSP1), business as usual (SSP2) and increased nationalism (SSP3), all have similar medium increases in phosphorus application to each other, however, it is the sustainable pathway (SSP1) which produces both the highest economic growth and lowest phosphorus use: making it the most desirable pathway. Under the business as usual scenario (SSP2) – which represents a middling pathway – Sub Saharan Africa will require approximately 440 million tonnes of phosphate rock between 2020 and 2050 with 74 % used in just 10 nations. Such an amount is well within current reserve estimates meaning that depletion will not be the limiting factor for increasing agricultural production. We project the total cost of phosphate rock as a raw material to be US\$ 50 ± 15 billion, however, the cost to African farmers is likely to be much higher – due to additional costs (e.g. taxation, transport etc.). Nevertheless, we have demonstrated that at a national level, despite early increases in expenditure on phosphate rock relative to GDP, the longer trajectory is for a decrease in expenditure relative to GDP as economies grow. A low phosphate rock expenditure relative to GDP is achieved only if agricultural production becomes reliable and national wealth increases.

AND

Line 454 (Conclusion) - This work has demonstrated that, regardless of which socioeconomic pathway Sub-Saharan Africa embarks upon there will be large increases in fertiliser application to 2050. The lowest increases in application rates, 310 %, will be under an increasingly unequal pathway (SSP4) and is because fewer people will become wealthy meaning few will develop expensive, resource intensive diets. The highest increase in phosphorus use (620 %), occurs when economic growth is stimulated by the extractive industries (SSP5), the other three pathways: sustainable (SSP1), business as usual (SSP2) and increased nationalism (SSP3), all have similar medium increases in phosphorus application to each other. Of these, SSP1 has the highest economic growth and lowest phosphorus use and hence is the most desirable pathway. The business as usual scenario, SSP2, represents a middling pathway. Here, between 2020 and 2050, Sub Saharan Africa will require approximately 440 million tonnes of phosphate rock with 74 % used in just 10 nations. Increases in phosphorus uptake (i.e. yield) will be driven by phosphorus application with the efficiency of phosphorus use largely dictated by intrinsic soil properties. The total cost of phosphate rock as a raw material under SSP2 will be US\$ 50 ± 15 billion but the cost to Sub-Saharan African farmers for phosphate fertiliser at the farmgate is likely to be much higher – due to additional costs (e.g. taxation, transport etc.). We have demonstrated however that despite early increases in expenditure on phosphate rock relative to GDP initially, in the longer term the ratio of expenditure to GDP should decrease as GDP increases with more reliable agricultural production.

We have rephrased the aims to try and improve the clarity of the scope of this study.

Line 203 (Introduction) - In this article we address the question of “*Can sub-Saharan Africa feed itself?*” through the lens of phosphate as a limiting resource. We assess: 1) how much phosphate will be required in Sub-Saharan Africa to 2050 under different pathways; 2) how uptake, application rates will vary in Sub-

Saharan Africa under a business as usual scenario (SSP2) 3) the cost of phosphorus under SSP2; and 4) if any Sub-Saharan African countries are expected to reap the hysteretic benefits seen in Western Nations.

Additionally, we updated the text to reflect our new modelling results:

Line 248 (Results and Discussion) - 2.3) Spatial & National Variations Phosphorus Futures Under SSP2 (Business as Usual)

We assess SSP2, the business as usual scenario, in more detail to derive spatial and country specific information about the future phosphate projections ^{35–37}.

2.3.1 Phosphate uptake and application by spatial distribution under SSP2

Our results show that in 2020 elemental phosphorus uptake in Sub-Saharan Africa was mostly restricted to 7 kg/ha or lower across most regions (Figure 2A). Elemental phosphorus application had a maximum of about 7 kg/ha but most of the continent had considerably lower application rates (Figure 2C). By 2050 we project these uptake rates will have increased dramatically to as high as 15 kg/ha in Benin and Ghana, and approximately 12 kg/ha over large parts of Cameroon, Gabon, Ethiopia, Nigeria, Republic of Congo, South Africa and Zambia (Figure 2B). To support this, most of these nations will have to increase application rates to as high as 15 kg/ha (Figure 2D).

2.3.2 Phosphate rock usage and expenditure under SSP2

We converted elemental phosphorus into phosphate rock (70 % BPL \approx 32 % P₂O₅) and projected that, under SSP2, Sub-Saharan Africa will have consumed approximately 440 million tonnes between 2020 and 2050. Over 74 % of this will be required in just 10 countries: Nigeria 24 %; Ethiopia 12 %; South Africa 7 %; Ghana 7 %; D. R. Congo 6 %; Côte d' Ivoire 5 %; Cameroon 5 %; Tanzania 3 %; Benin 3 % and Mali 3 % (Figure 3A). Between 2020 and 2050 we project the four highest phosphate consuming nations to consume 50 % of the phosphate rock used and we assess these in more detail.

The increases in application are not evenly distributed. Based on a baseline of 2010 to 2020, we project that Nigeria, Ethiopia, South Africa and Ghana will have increases in phosphate uses of 934 %, 375 %, 148 % and 130 %, respectively, over the 30 years to 2050. Between 2010 and 2020, Nigeria, Ethiopia, South Africa and Ghana consumed an average of 0.46 ± 0.26 , 0.64 ± 0.21 , 0.61 ± 0.05 and 0.10 ± 0.12 million tonnes of phosphate rock per year and by 2050, under SSP2, we project this consumption to be 4.3, 2.8, 0.9 and 1.3 million tonnes of phosphate rock, respectively, (Figure 3B & C).

AND

Line 398 (Results and Discussion) – 2.3.3 Phosphate hysteretic crop uptake effect under SSP2

Under specific scenarios phosphorus uptake (and hence yield) can increase despite declining phosphorus input – this is called the hysteretic crop uptake effect. The hysteretic crop uptake effect is beneficial to regional agriculture since application rates becomes less important. At a continental level Western Europe has benefitted from this effect since 1970, Latin America and Asia are expected to benefit in the coming decades, however, Africa requires substantial increases in phosphate fertiliser application ⁶. The hysteretic crop uptake effect is caused when part of the applied phosphorus is adsorbed by soil and is gradually released over time. The adsorption and release rates of phosphorus in soils depends on the soil chemistry, mineralogy and microbiological conditions. The difference in adsorption and release rates of phosphorus is the a “hysteretic effect”.

Our results demonstrate that historically, Nigeria, South Africa, Democratic Republic of Congo and Cameroon have experienced the hysteretic crop uptake effect (Figure 4). However, no studied country is projected to experience the hysteretic effect under SSP2. This is due local variations in phosphorus flows within the soil which we discuss in detail in the supplementary information. Most importantly, this means that all nations in this study will rely on phosphorus application – with associated costs – as part of the intensification process.

New Figures

All figures have been updated and are provided here.

Total Annual Elemental Phosphorus Required for Sub-Saharan Africa 1950 to 2050

Figure 1 Total elemental phosphorus application (1000 tonnes) in Sub-Saharan Africa from 1950 to 2050, with 2020 to 2050 projected under Shared Socioeconomic Pathways⁵⁷ within IMAGE3.0 framework⁵⁸.

Figure 2 Maps of Sub-Saharan Africa showing phosphorus uptake rates (kg/ha) for 2020 (A) & 2050 (B) and phosphorus fertiliser application (kg/ha) rates for 2020 (C) & 2050 (D) under SSP2.

Figure 3 For SSP2 (A) Total phosphorus required (1000 tonnes) and percentage breakdown for Sub-Saharan Africa between 2020 and 2050; (B) historic and projected phosphate rock use (1000 tonnes) 1950 to 2050 in Nigeria, Ethiopia, South Africa and Ghana; (C) historic and projected phosphate rock costs (million US\$) 1950 to 2050 in Nigeria, Ethiopia, South Africa and Ghana; and (D) historic and projected phosphate rock expenditure relative to GDP (USD \$/ USD \$, unitless) 1950 to 2050 in Nigeria, Ethiopia, South Africa and Ghana. Phosphorus and phosphate rock data are calculated by this project, phosphate historic rock price and GDP are provided by the World Bank ^{25,45} and projected GDP ⁴⁴.

Figure 4 Total phosphorus inputs (fertiliser and manure, kg/ha/yr) versus total phosphorus uptake (kg/ha/yr) for the period 1970 to 2050 for the 10 highest phosphorus consuming countries in Sub-Saharan Africa under SSP2.

REVIEWER COMMENTS

Reviewer #1 (Remarks to the Author):

The authors have made some major revisions in response to Reviewer comments. Many of these have clarified my concerns (e.g. including multiple SSP scenarios, and resolving the 'farm gate' fertiliser price issue through caveats and removing the VCR component). The new figures are great – especially Figure 1 with a range of SSP phosphorus scenarios. However the manuscript is still not of a high quality as it stands as there are still some important residual aspects that have not been dealt with and would still need to be addressed. The new text, while helpful, also would need to be brought up to a higher quality, as outlined below.

AGRICULTURAL PRODUCTION VS FOOD SECURITY:

The manuscript addresses the contribution that changing agricultural production can contribute to food security and "Africa feeding itself". As mentioned in the first review round, while sustainable intensification (and sustainable agricultural production) is key, they alone do not equate to food security. There are MANY additional fundamental dimensions that are also required to achieve food security, many of these 'off-farm' (e.g. particularly the access to and affordability of food, significant shift to healthy and balanced diets, efficient post-farm value chains (including reduced post-harvest losses), farmer access to credit, rural youth training and employment, etc).

Many of these dimensions have important implications for phosphorus demand in Sub-Saharan Africa (and globally). E.g. changing diets towards healthy/balanced diets means some major shifts in growing different foods/crops, possibly different land-uses and geographical locations which all result in different phosphorus demands. As mentioned in the previous comments, the authors soil-phosphorus expertise could make fantastic contributions to understanding these implications, if they teamed up with experts in planetary health/dietary shifts. I understand that might be outside the scope of this present study, but it is the larger value I think these authors could provide to the field. Regardless, not explicitly acknowledging these linkages in the present manuscript is problematic for the findings so needs to at least be discussed in a meaningful way.

The author's response to my first round of comments on this point acknowledged this: "Agreed, thank you, done. We did conflate "food security" with "agricultural production". We have corrected the terminology throughout". However the two are still somewhat incorrectly conflated and confused in the text. This is misleading in itself, and also misleads the reader regarding what the phosphorus results do and don't represent. For example:

- o The title "Can Sub-Saharan Africa feed itself?" will need to better reflect the shift in focus from food security to agriculture production, as they are not the same thing, as has been discussed by reviewers, and addressed by authors.

- o Line 276- "dietary requirements" – does this refer to the calorie intake? I assume it does not refer to actual balanced and healthy diets, which requires a shift in what is grown and where. As discussed in the initial review comments. If the former, this needs to be made more transparent and clear.

- o It is great to see the authors have included multiple SSP scenarios for the first analysis in the revision (Figure 1 plus the supporting text/discussion). What is still somewhat unclear is the assumptions behind these key SSP scenarios as they relate to/affect agriculture, food security and phosphorus. Dietary change is the most obvious, as per above. The authors have now mentioned "diets" but it is unclear if they are referring to quantity (total calorific changes kcal/capita), or actual quality shifts in say vegetable consumption, dairy, legumes etc etc that have an impact on

land-use and phosphorus demand (I don't think SSP1 Sustainable Pathway includes explicit modification of land-use in response to shifts towards a 'sustainable' planetary diet, except for reduced meat intake. And what about the impact of reduced food waste on reduced calorific food demand due to less waste?). It would be helpful to include the relevant SSP assumptions/exclusions in the Supplementary Material, since they are crucial to the analysis. You could cross-reference this from your para on IMAGE lines 202-267 (and the para before) where you have indeed outlined some of the assumptions.

o I provided a range of suggested references to make it easier for the authors with respect to dietary shifts (in the first round of Review comments). The authors have referred to some of these, however there are also many other recent 2020 resources in relation to non-agricultural food security challenges, e.g.

- FAO 2020 - <http://www.fao.org/publications/sofi/2020/en/>

- CERES2030 https://ceres2030.org/shorthand_story/donors-must-double-aid-to-end-hunger-and-spend-it-wisely/

EAT-Lancet study

I mentioned EAT-Lancet study a few times, because I think it is highly relevant here both as a 'data source' and also to compare findings to since the EAT-Lancet study did some similar work to your analysis (Springmann et al 2018 (NATURE) sent earlier).

Springmann et al also project P demand out to 2050 based on the SSPs, including BAU and healthy diets scenarios. How do your figures compare to theirs? (did you use different assumptions?). What is the novel contribution of your study compared to the EAT-Lancet study? (e.g. is your soil model more nuanced?). So in your discussion, you would need to compare your results to these.

PHOSPHATE ROCK VS FERTILISERS:

The authors have perhaps misunderstood my (and possibly Reviewer #3) comments regarding the use of "phosphate rock" expenditure rather than "fertilisers". I was not referring to the small direct application of crushed phosphate rock on-farm. I was referring to the fact that most SSA countries would not import phosphate rock – they would import phosphorus-containing fertiliser products (DAP etc). The price of phosphate rock doesn't directly translate to the cost of fertilisers due to other raw materials in fertilisers and dynamics at play (this is a separate/additional issue to the farm-gate price of fertiliser issue that I raised last time, which the authors have now referred to in a caveat). This former issue hasn't been reconciled or acknowledged as a caveat to the analysis. What is the relationship between the price of phosphate rock and the price of fertilisers, and what are the implications of using phosphate rock as a proxy for fertiliser expenditure?

OTHER COMMENTS ON NEW TEXT:

The new text, while substantial and appreciated (and supported by a detailed 20 page response to reviewers), now contains new quality concerns that would need to be addressed:

- The new text appears to have been written in haste and not proofed by the authors before resubmitting. Grammar, spelling etc need to be reviewed/edited the new text – e.g. ('calory' spelling, poor sentence structure, and errors, etc).

-Lines 33-34: This new sentence needs revision:

o the second half of the sentence is currently plagiarised from the original authors (modify sentence)

o Even without plagiarism, it needs to be directly referenced to the authors

o It needs a qualifier like "due to population growth and changing diets" at the end of the sentence: "The 21st century faces a "food gap": the estimated 70% difference between the crop calories available in 2006 and expected calorie demand in 2050"

- Lines 89: "Increased AND EFFICIENT use of fertiliser"
- Line 180 – this wont be immediately obvious to non-phosphorus experts that you are talking about "recycling elemental phosphorus from organic wastes like food waste and wastewater along the phosphorus value chain"
- Line 195 - suggest change "controlled" to "heavily influenced"
- Line 205 – change "human" to "socio-economic and socio-technical" factors.
- Line 280 – The use of SSA, Africa and "Continental Africa" is not consistent across the manuscript, but it needs to be as the results will be very different for different regions. E.g. this line's heading uses "Continental Africa", yet the first sentence refers to "Sub Saharan Africa" in the results?
- Line 341 – "Under BUSINESS-AS-USUAL (SSP2)"
- Line 467-468 – "which also saw an increase in phosphorus use in the country" - it's hard to see the dynamics clearly in Figure 3B, but it actually looks a bit like the phosphorus use plateaued/stagnated around 2008, then increased after?
- Figure 3:
 - o The right scale on Figure 3A looks like it is the axis heading for Figure 3B.
 - o Figure 3B – Y axis heading "Phosphate Rock Applied" does now sound like you're referring to direct application of crushed rock. Change to something more accurate like "phosphate rock used in fertiliser application"
- Line 513 – suggest adding "also known as legacy phosphorus". Or clarifying how these terms relate, for the wider audience.
- Conclusion – there are a few large value judgments here:
 - o Line 709 – "and hence is the most desirable pathway" - this is a grand statement/assumption. There are many other factors besides GDP and P use at play to make something the most desirable pathway. E.g. this could be at the expense of significant environmental pollution. Best to say something like "and hence is the most desirable pathway when considering only GDP and phosphorus use".
 - o "The business as usual scenario SSP2 represents a middling pathway" – this can be easily wrongly interpreted as a middle-of-the-road pathway for phosphorus, and hence sounds loaded. The previous sentence refers to a 'desirable' pathway, so 'middle' implies medium desirability, which is not necessarily for food security or phosphorus. Much of the food security literature points out business as usual is indeed UNDESIRABLE due to the health and environmental costs that now outweigh the benefits of agriculture.
 - o These are referring to future predictions under very specific modelling scenarios, so you need to avoid such decisive terms like "will be" and "will require". I think I mentioned this in the first round of Reviewer comments. These need to be qualified under the scenario, e.g. "will likely be", or "will require more under this scenario".

Reviewer #2 (Remarks to the Author):

Dear authors

I am happy with the revised version. Just minor review before going forward:

Line 104 - there is a mistake in the second price number;
Line 205 and others referring to fig3 - Error, please check;
Line 283 - should link the term "hysteretic effect" to the well-defined term "legacy P" shown in many publications worldwide.

Best regards

Paulo Pavinato

Reviewer #3 (Remarks to the Author):

Thank you for making the changes you have made to this manuscript, which is much improved.

The key issue now concerns proofreading and tidying up the English. The comma, in particular, is poorly used and there are written errors and inconsistencies - with the following examples:

- 1) First sentence of abstract needs to be rewritten
- 2) Be consistent with the use of calory or calorie as the singular of calories
- 3) Line 76-77 what do you mean: 'for the duration'? This is a completely unspecified term
- 4) Lines 91-96 need careful proof reading
- 5) Line 155: is this correct: "800 % from US\$ 50 to US\$ 50 430"?

In terms of the technical content, I appreciate that SSP5 relates to 'fossil fuelled development', but the emphasis is on fossil fuels for energy rather than mined materials. I think you should make it clear that your interpretation of SSP5 specifically encompasses mined raw materials.

However, this does lead to your statement in the Conclusions that "economic growth is stimulated by the extractive industries (SSP5)". This aspect is not discussed specifically earlier, and so is a new idea that enters the conclusions. I think it should be discussed, as one of the threads running through the first draft, and still in the second draft, is an apparent lack of understanding of how the mining industry works, and of how the term 'reserve' is very specifically defined in regulations to provide a figure that investors can trust - to a common standard internationally. 'Reserves' are an entirely artificial construct that have no relationship to the rocks in the ground, other than the figure reported has been proved by robust investigation of those rocks. So the reserves figures for many commodities in the USGS reports that you cite are constant from year to year, simply because the pace of their definition equals the pace of their extraction. For an investor, that is a good place to be.

The extractive industries do not stimulate economic growth by themselves. They only exist because investors are convinced that there is demand for their products, so it is the other way round: society stimulates demand for extraction. Just look at the current lithium 'gold rush' as an example of how that works as an SSP5 driver designed to counter the fossil fuels. You can see this relationship with investors in the York Potash project, where \$billions have been raised from investors to build a mine on the basis of long-term purchase agreements for the fertiliser minerals produced by the mine once it is built. Those agreements give confidence that the investors will get their money back.

RESPONSE TO REVIEWERS' COMMENTS

Reviewer 1:

The authors have made some major revisions in response to Reviewer comments. Many of these have clarified my concerns (e.g. including multiple SSP scenarios, and resolving the 'farm gate' fertiliser price issue through caveats and removing the VCR component). The new figures are great – especially Figure 1 with a range of SSP phosphorus scenarios. However the manuscript is still not of a high quality as it stands as there are still some important residual aspects that have not been dealt with and would still need to be addressed. The new text, while helpful, also would need to be brought up to a higher quality, as outlined below.

R1.1 We are glad that the reviewer's concerns have been partially allayed with 1) projection of four new SSPs; 2) caveats on farm-gate price and 3) removal of VCR. We will take no further actions on these aspects, but we recognise that there are further areas of improvement required and are grateful for the opportunity to address these.

AGRICULTURAL PRODUCTION VS FOOD SECURITY:

The manuscript addresses the contribution that changing agricultural production can contribute to food security and "Africa feeding itself". As mentioned in the first review round, while sustainable intensification (and sustainable agricultural production) is key, they alone do not equate to food security. There are MANY additional fundamental dimensions that are also required to achieve food security, many of these 'off-farm' (e.g. particularly the access to and affordability of food, significant shift to healthy and balanced diets, efficient post-farm value chains (including reduced post-harvest losses), farmer access to credit, rural youth training and employment, etc).

Many of these dimensions have important implications for phosphorus demand in Sub-Saharan Africa (and globally). E.g. changing diets towards healthy/balanced diets means some major shifts in growing different foods/crops, possibly different land-uses and geographical locations which all result in different phosphorus demands. As mentioned in the previous comments, the authors soil-phosphorus expertise could make fantastic contributions to understanding these implications, if they teamed up with experts in planetary health/dietary shifts. I understand that might be outside the scope of this present study, but it is the larger value I think these authors could provide to the field. Regardless, not explicitly acknowledging these linkages in the present manuscript is problematic for the findings so needs to at least be discussed in a meaningful way.

R1.2 We are glad the reviewer has recognised the changes already implemented. We recognise that food security is a complex and multi-faceted issue and that there is not the space to explore every avenue of this topic in an article for *Nature Communications*' length. Our approach is to be disciplined in sticking to the topic of the role of phosphorus in Sub-Saharan agricultural production to 2050 within the SSP framework. To improve the clarity of this we have made further revisions to the text to address the concerns raised here: 1) we focus this on the role of phosphorus in SSA agricultural production within the SSP framework and remove references to food security; 2) we add meaningful discussion regarding diets with references previous studies; and 3) add greater description of the socio-economic components which constrain the model.

Specifically, we have added the following text:

Line 223 (Introduction) – IMAGE 3.2²⁶ predicts that by 2050 the lowest Sub-Saharan Africa population growth is likely to be under sustainable SSP1 and fossil fuelled SSP5 both of which have 1.6 billion people. The highest population growth is likely to be under the increased nationalism of SSP3 and inequality of SSP4 both of which have populations of 2 billion people projected. The middle of the road SSP2 reaches 2050 with 1.8 billion people

(**Error! Reference source not found.A**). GDP is broadly the inverse of population with SSP5 highest, followed by SSP1, SSP2, SSP3 and SSP4 at the lowest; these have per-capita values of US\$ 7300, US\$ 5300, US\$ 3000, US\$ 2000 and US\$ 2000, respectively (**Error! Reference source not found.B**).

Food production follows GDP. Meat, milk and crop production are highest under fossil fuelled SSP5 and the lowest under the unequal SSP4. Despite being the lowest SSP4 still requires a 2.6-fold increase total production from 2015 levels (**Error! Reference source not found.C, D & E**). This pattern emphasises that it is the wealth of the population as well as the total number of people that drive's agricultural production demand.

By 2050, the greatest yields are likely to be under both the fossil fuelled SSP5 and the sustainable SSP1 (both 2.45 Mg dry matter/ ha /yr) whilst the lowest yields are under the unequal SSP4 (1.94 Mg dry matter/ ha /yr, **Error! Reference source not found.F**). Together this means that the production area is likely to be highest under SSP5 (470 Mha) followed by SSP3 (460 Mha), SSP2 (420 Mha), SSP4 (410 Mha) and lowest in SSP1 (380 Mha) (**Error! Reference source not found.G**). SSP5 was originally envisaged as “fossil fuelled growth”²³, however, we broaden the term to mean environmentally exploitative (e.g. expanded agricultural lands, high water use, high greenhouse gas emissions) within the context of high GDP growth²².

Whilst the volume and yield of crops vary under the different scenarios the crop production ratios (i.e. production of crop A compared to crop B) stay relatively constant in the IMAGE 3.2 scenarios. By 2050 “tropical roots and tubers” are projected to be the most abundant crop type in Sub-Saharan Africa followed by “maize”, “tropical cereals” and “tropical oil crops” (see supplementary information).

AND

Line 283 (Results) – These results are driven by food production requirements. For example, fossil fuelled SSP5, has the highest level of GDP growth coupled with high crop, meat and milk production (**Error! Reference source not found.**) and therefore the highest amount of required phosphorus. By contrast, the unequal society of SSP4 has limited GDP growth coupled to limited crop, meat and dairy production (**Error! Reference source not found.**) and hence the lowest phosphorus requirements (**Error! Reference source not found.**). Our production assessment concurs with an earlier global dietary assessment which highlights that reduction in meat consumption as a mechanism for reducing phosphorus demand³.

However, land use change also effects crop production levels and related phosphorus requirements. This is most clearly demonstrated by the difference in phosphorus demand between the regional rivalries of SSP3 and the unequal SSP4 (**Error! Reference source not found.**). Both have similarly high rates of population growth and low rates of GDP (**Error! Reference source not found.B**), yet, SSP4 has the lowest phosphorus requirements whilst SSP3 has relatively high phosphorus requirements (**Error! Reference source not found.**). This is caused by the expansion of croplands under SSP3 which enables meat and milk production to increase at a greater rate than the SSP4 despite similar GDP growth (**Error! Reference source not found.B&G**). This highlights the necessity of targeted, efficient phosphorus with good land management use to prevent unnecessary expansion^{4,5}.

Good land management, efficient resource use and technological change are modelled under the sustainable pathway of SSP1²³. Here population growth is low and GDP comparatively high (although not as high as environmentally exploitative SSP5). Under SSP1, meat, milk and crop production increase along a middling

pathway: greater than unequal SSP4 but less than environmentally exploitative SSP5 and similar to middle of the road SSP2 and the regional rivalries of SSP3 (**Error! Reference source not found.**C, D and E). However, this pathway has a high yield rate (**Error! Reference source not found.**F), albeit slightly less than SSP5, and has the least expansion of croplands ²⁶. By targeting phosphorus efficiently high crop yields along with high meat and dairy production are sustained whilst limiting the amount of phosphorus used and the expansion of croplands.

This indicates that within the context of GDP and phosphorus the sustainable pathway SSP1 is the most favourable since it has comparatively high levels of GDP but requires relatively low amounts of phosphorus and low increases in agricultural lands to attain this. The unequal society of SSP4 also contains low phosphorus requirements but this is because GDP does not increase which leads to a lack of meat and milk production. The greatest rise in GDP is in environmentally exploitative SSP5 but this requires large amounts of phosphorus and large expansion of crop lands. Within the GDP-phosphorus context the middle of the road of SSP2 and regional rivalries of SSP3 are arguably the least beneficial outcomes with low rises in GDP coupled with large increases in both phosphorus and agricultural lands. Therefore, within the GDP- phosphorus context middle of the road SSP2 does not provide a favourable future.

The author's response to my first round of comments on this point acknowledged this: "Agreed, thank you, done. We did conflate "food security" with "agricultural production". We have corrected the terminology throughout". However the two are still somewhat incorrectly conflated and confused in the text. This is misleading in itself, and also misleads the reader regarding what the phosphorus results do and don't represent. For example:

The title "Can Sub-Saharan Africa feed itself?" will need to better reflect the shift in focus from food security to agriculture production, as they are not the same thing, as has been discussed by reviewers, and addressed by authors.

R1.3 We accept that the title must change to reflect the agricultural production we have changed the title to "**The phosphorus pathways of Sub-Saharan Africa's food production future**".

Line 276- "dietary requirements" – does this refer to the calorie intake? I assume it does not refer to actual balanced and healthy diets, which requires a shift in what is grown and where. As discussed in the initial review comments. If the former, this needs to be made more transparent and clear.

R1.4 We have changed this to "calory intake" and elsewhere the changes made a R1.2 have clarified this further.

It is great to see the authors have included multiple SSP scenarios for the first analysis in the revision (Figure 1 plus the supporting text/discussion). What is still somewhat unclear is the assumptions behind these key SSP scenarios as they relate to/affect agriculture, food security and phosphorus. Dietary change is the most obvious, as per above. The authors have now mentioned "diets" but it is unclear if they are referring to quantity (total calorific changes kcal/capita), or actual quality shifts in say vegetable consumption, dairy, legumes etc etc that have an impact on land-use and phosphorus demand (I don't think SSP1 Sustainable Pathway includes explicit modification of land-use in response to shifts towards a 'sustainable' planetary diet, except for reduced meat intake. And what about the impact of reduced food waste on reduced calorific food demand due to less waste?). It would be helpful to include the relevant SSP assumptions/exclusions in the Supplementary Material, since they are crucial to the analysis. You could cross-reference this from your para on IMAGE lines 202-267 (and the para before) where you have indeed outlined some of the assumptions.

R1.5 The two additions for R1.2 have addressed these concerns by describing relative crop, meat, and milk production at the continental scale for SSP1 – 5 along with other drivers. In the supplementary information we

have provided edited versions of the original SSP narratives of O'Neill et al. (2017 – text not provided in this document) and discussion on the changes in crop type under the different scenarios.

Line 25 (Supplementary information) - The different shared socio-economic pathways within IMAGE 3.2 have contrasting total crop-productions which we discuss in the main text. However, the model has 16 different crop types within it: Maize, Oil – palm fruit, other non-food (e.g. luxury, spices), other temperate cereals, plant based fibres, pulses, rice, soy beans, sugar crops, temperate oilcrops, temperate roots and tubers, tropical cereals, tropical oilcrops, tropical roots and tubers, vegetables and fruits, and wheat. These follow subtly different pathways but by 2050 IMAGE 3.2 projects only small variations between the crop ratios for the different pathways. However, in all SSPs three crop types, tropical roots and tubers, maize, and tropical cereals dominate Sub-Saharan Africa crop production making up $29 \pm 1 \%$, $14 \pm 1 \%$ and $10 \pm 1 \%$, respectively, with all other crops consisting of the other $48 \pm 2 \%$.

I provided a range of suggested references to make it easier for the authors with respect to dietary shifts (in the first round of Review comments). The authors have referred to some of these, however there are also many other recent 2020 resources in relation to non-agricultural food security challenges, e.g.

- FAO 2020 - <http://www.fao.org/publications/sofi/2020/en/>

- CERES2030 https://ceres2030.org/shorthand_story/donors-must-double-aid-to-end-hunger-and-spend-it-wisely/

R1.6 We are grateful to the reviewer for providing us with the additional references: we included all the references except for Robinson et al., (2015) which we omitted since we use IMAGE rather than IMPACT. In these revisions we have given more prominence to the Springmann work using it to set the context of our work rather than bringing in the findings later on only. We have also provided much greater focus on the IMAGE 3.2 findings to set the socio-economic framework within which we predict.

We have provided five additional references to the previous draft but have focussed these on phosphorus.

- Mogollón, J. M. *et al.* More efficient phosphorus use can avoid cropland expansion. *Nat. Food* 2021 1–10 (2021). doi:10.1038/s43016-021-00303-y
- Mew, M. C. Phosphate rock costs, prices and resources interaction. *Sci. Total Environ.* **542**, 1008–1012 (2016).
- UNCTAD World Bank and Equitable Maritime Consulting. *Global Transport Costs Dataset for International Trade: Trade by all modes of transport – 2016 – output variables – transport costs, transport costs to FOB value, transport costs per unit, transport costs per unit per 10 000 km.* (2020).
- The World Bank. *International Monetary Fund, Government Finance Statistics Yearbook and data files: Taxes on international trade (% of revenue): GC.TAX.INTT.RV.ZS.* (2021).
- Federal Government of Nigeria. *Nigeria Single Window Trade: CET Tariff - Act No. 4.*

EAT-Lancet study

I mentioned EAT-Lancet study a few times, because I think it is highly relevant here both as a 'data source' and also to compare findings to since the EAT-Lancet study did some similar work to your analysis (Springmann et al 2018 (NATURE) sent earlier).

Springmann et al also project P demand out to 2050 based on the SSPs, including BAU and healthy diets scenarios. How do you figures compare to theirs? (did you use different assumptions?). What is the novel

contribution of your study compared to the EAT-Lancet study? (e.g. is your soil model more nuanced?). So in your discussion, you would need to compare your results to these.

R1.7 Springmann's work is highly pertinent to our own. The main difference between our work and Springmann's is that our work focusses on the pre-production process whilst Springman focuses on post-production. In other words, whilst Springmann addresses the consequence of phosphorus use they don't provide spatially explicit estimates of the amount required or set this into a GDP-geological resource framework.

We have added in text discussing their findings in their own right in the introduction as well as in relation to our own findings in results and discussion (the latter are in R1.3).

Line 52 (Introduction) - Today the challenge is how to meet calory demand whilst minimising ecological and environmental damage. A recent analysis highlighted three likely dietary change scenarios which provide health benefits: reduction of animal-source foods, improved energy balance & weight levels and a balanced, more plant based flexitarian diet. These also reduce environmental impact of food for several key markers: greenhouse gas emissions, nitrogen use, freshwater use, croplands used and phosphorus use. However, in low income countries, unlike more developed regions, flexitarian diets are likely to increase environmental and ecological pressure using 15 %, 25 % and 7 % more cropland, freshwater and phosphorus, respectively ³. Therefore ambitious technological change (e.g. improving agricultural yields, fertiliser efficacy, targeted use of fertiliser, livestock feed efficiency and improved management) is critical to reducing the environmental impact of food production ^{4,5}.

PHOSPHATE ROCK VS FERTILISERS:

The authors have perhaps misunderstood my (and possibly Reviewer #3) comments regarding the use of "phosphate rock" expenditure rather than "fertilisers". I was not referring to the small direct application of crushed phosphate rock on-farm. I was referring to the fact that most SSA countries would not import phosphate rock – they would import phosphorus-containing fertiliser products (DAP etc). The price of phosphate rock doesn't directly translate to the cost of fertilisers due to other raw materials in fertilisers and dynamics at play (this is a separate/additional issue to the farm-gate price of fertiliser issue that I raised last time, which the authors have now referred to in a caveat). This former issue hasn't been reconciled or acknowledged as a caveat to the analysis. What is the relationship between the price of phosphate rock and the price of fertilisers, and what are the implications of using phosphate rock as a proxy for fertiliser expenditure?

R1.7 To address these points we have made some additional calculations and added some extra commentary.

Line 429 (Results) - As mentioned, phosphate rock is not the only cost within fertiliser. Production, transportation, taxation and profit margins all contribute to the final product price ⁴². We provide estimates of these additional costs to the phosphate rock price at the point of import for countries which import fertiliser or at the point of production for self-producing countries. We calculated this for the years 2010, 2030 and 2050 with the future phosphate price, again, assumed to be the mean price between 2010 and 2020. Production costs were assumed to be US\$ 38 per tonne ⁴⁵. Transportation is the cost of transporting phosphate fertiliser product from Morocco to the country of use in 2016 ⁴⁶ (except South Africa which is assumed to be self-producing ¹¹). We do not include internal transportation costs. Taxation is calculated at the current rate of import tax for each country in 2020 ^{47,48}. We did not include any internal taxes and for this reason South Africa is assumed not to have an import tax due to self-production ¹¹. Profit margins were set at 7.5 % with no other informal costs assumed ⁴².

We emphasise that the values we provide here are for illustrative purposes only and not predictions. Factors such as a change in government, policy, war, natural disaster, technology and market forces would alter these numbers greatly. Nevertheless, the results highlight that such additional costs differed between the nations. South Africa has the lowest additional costs due to being self-reliant, this means that 69 % of the cost was phosphate rock. However, by contrast Ethiopia has a high rate of import tax and transportation costs meaning that only 53 % of the fertiliser price was phosphate rock. For Nigeria and Ghana phosphate rock consists of 59 % and 57 % of the final phosphate fertiliser price, respectively. For Nigeria, this is mostly due to transportation costs whilst for Ghana this is a mix of transportation and taxation.

Line 511 (Methods) - Additional prices, in phosphate fertiliser other than phosphate rock were assumed to be production, international transportation, taxation and profit margins ⁴². We calculated these values for illustrative purposes only – particularly for the future years – with the numbers based on contemporary additional prices. We recognise that factors such as a change in government, policy, war, natural disaster, technology and market forces could alter these numbers greatly and as such they should not be considered projections.

Regardless, a 2016 review by Mew indicated production costs are US\$ 38 per tonne ⁴⁵. Morocco is the dominant phosphate supplier to Sub-Saharan African countries so we assumed transport costs from Morocco to the country of use. Transport costs were calculated by multiplying the sum of the phosphate rock and production costs by the FOB to transport costs ratio provided by the UNCTAD Global Transport Costs Dataset for International Trade for 2016. UNCTAD does not supply values for Ethiopia so a simple average of Uganda and Tanzania are used instead ⁴⁶. We assume South Africa to be self-producing so do not include a transport cost. Commodity import tax is supplied from the World Bank for Ghana and Ethiopia ⁴⁷ and from the Federal Government of Nigeria ⁴⁸ no import tax is assumed for South Africa which is assumed to be self-producing ¹¹. Tax was calculated on the sum of phosphate rock, transport and production costs. Finally, a profit margin of 7.5 % ⁴² was assumed in all countries and calculated on the sum of phosphate rock, transport, production and taxation costs. The transport values and tax values used are provided in **Error! Not a valid bookmark self-reference.** based on values from 2016 and 2020 respectively.

Table 1 Values used to calculate transport and taxation costs in total fertiliser costs.

	2016 Transport costs to FOB %			Ref.	Transport cost to FOB % used	2020	Ref.
	Mineral Chemical Fertilisers	DAP	ADP			Taxation %	
Nigeria	20.00	14.00	ND	⁴⁶	17.00	0	⁴⁸
Ethiopia	ND	ND	ND	ND	13.03	15.2	⁴⁷
South Africa	ND	ND	13.63	⁴⁶	0 *	3.4 *	⁴⁷
Ghana	10.63	ND	ND	⁴⁶	10.63	10.1	⁴⁷

Tanzania	ND	13.27	ND	⁴⁶	NA	NA	NA
Uganda	ND	ND	12.78	⁴⁶	NA	NA	NA

ND = no data supplied by source; NA = not applicable for this study; * a value of 0 is used assuming South Africa is self-sufficient in phosphate rock ¹¹.

OTHER COMMENTS ON NEW TEXT:

The new text, while substantial and appreciated (and supported by a detailed 20 page response to reviewers), now contains new quality concerns that would need to be addressed:

- The new text appears to have been written in haste and not proofed by the authors before resubmitting. Grammar, spelling etc need to be reviewed/edited the new text – e.g. ('calory' spelling, poor sentence structure, and errors, etc).

- Lines 33-34: This new sentence needs revision:

- o the second half of the sentence is currently plagiarised from the original authors (modify sentence)

- o Even without plagiarism, it needs to be directly referenced to the authors

- o It needs a qualifier like “due to population growth and changing diets” at the end of the sentence: “The 21st century faces a “food gap”: the estimated 70% difference between the crop calories available in 2006 and expected calorie demand in 2050”

R1.8 We have corrected this and it now reads “The 21st century faces a “food gap”: the estimated 70 % difference between the amount of crop calories produced in 2006 and the expected amount of crop calories required by 2050 given population growth and global dietary change.” (**Line 44 – Introduction**).

- Lines 89: “Increased AND EFFICIENT use of fertiliser”

R1.9 We have added in “and efficient”.

- Line 180 – this wont be immediately obvious to non-phosphorus experts that you are talking about “recycling elemental phosphorus from organic wastes like food waste and wastewater along the phosphorus value chain”

R1.10 We have cut this line as part of our editing process we felt it less relevant than other content.

- Line 195 - suggest change “controlled” to “heavily influenced”

R1.11 Corrected.

- Line 205 – change “human” to “socio-economic and socio-technical” factors.

R1.12 Corrected (and at line 130).

- Line 280 – The use of SSA, Africa and “Continental Africa” is not consistent across the manuscript, but it needs to be as the results will be very different for different regions. E.g. this line’s heading uses “Continental Africa”, yet the first sentence refers to “Sub Saharan Africa” in the results?

R1.13 We have corrected this heading to “Continental-scale Sub-Saharan African Phosphorus Future under SSP1 to 5”

- Line 341 – “Under BUSINESS-AS-USUAL (SSP2)”

R1.14 Corrected, throughout.

- Line 467-468 – “which also saw an increase in phosphorus use in the country” - it’s hard to see the dynamics clearly in Figure 3B, but it actually looks a bit like the phosphorus use plateaued/stagnated around 2008, then increased after?

R1.15 This should have been Figure 3C, we have corrected this.

Figure 3:

The right scale on Figure 3A looks like it is the axis heading for Figure 3B.

Figure 3B – Y axis heading “Phosphate Rock Applied” does now sound like you’re referring to direct application of crushed rock. Change to something more accurate like “phosphate rock used in fertiliser application”

R1.16 Thank you for pointing this out, we have followed your suggestion

Line 513 – suggest adding “also known as legacy phosphorus”. Or clarifying how these terms relate, for the wider audience.

R1.17 Two reviewers have made this this suggestion. We have added the “however, no studied country is projected to experience the hysteretic effect under SSP2 due to a lack of sufficient residual (or legacy) phosphorus” (*Line 462 – Results*).

Conclusion – there are a few large value judgments here:

Line 709 – “and hence is the most desirable pathway” - this is a grand statement/assumption. There are many other factors besides GDP and P use at play to make something the most desirable pathway. E.g. this could be at the expense of significant environmental pollution. Best to say something like “and hence is the most desirable pathway when considering only GDP and phosphorus use”.

R1.18 Agreed, we have qualified this with “within the phosphorus-GDP context”.

“The business as usual scenario SSP2 represents a middling pathway” – this can be easily wrongly interpreted as a middle-of-the-road pathway for phosphorus, and hence sounds loaded. The previous sentence refers to a ‘desirable’ pathway, so ‘middle’ implies medium desirability, which is not necessarily for food security or phosphorus. Much of the food security literature points out business as usual is indeed UNDESIRABLE due to the health and environmental costs that now outweigh the benefits of agriculture.

R1.19 Agreed, in response to this we have removed subjective value statements where possible we have also highlighted (semi-quantitatively) the environmental damage of pathways such as SSP2.

Line 24 (Abstract) - The sustainable pathway (SSP1) has the highest economic growth for the lowest expansion of agricultural lands and phosphorus use whilst the middle of the road (SSP2) and increased nationalism pathways (SSP3) are middling but environmentally costly pathways for limited GDP growth.

Line 336 (Results) - The middle of the road scenario of SSP2 represents a middling, though not a favourable, pathway that we use to derive spatial and country specific information about the future phosphate projections ²²⁻²⁴.

Line 481 (Conclusion) – The other middling pathways SSP2 and 3 are not compromise pathways since they require greater increases of agricultural lands compared to SSP1.

Line 492 (Conclusion) – Nevertheless, due to the environmental costs of large expansion of agricultural lands couple with high levels of phosphorus use and moderate GDP increase, withing the GDP-phosphorus context business-as-usual SSP2 is not a favourable future.

These are referring to future predictions under very specific modelling scenarios, so you need to avoid such decisive terms like “will be” and “will require”. I think I mentioned this in the first round of Reviewer comments. These need to be qualified under the scenario, e.g. “will likely be”, or “will require more under this scenario”.

R1.20 Agreed, we have qualified these terms as suggested.

Reviewer 2:

Dear authors

I am happy with the revised version.

R2.1 We are glad you are happy with the manuscript and thank you for taking the time to review it twice.

Just minor review before going forward:

Line 104 - there is a mistake in the second price number;

R2.2 Yes, this should have been \$ 430. We have corrected this, thanks.

Line 205 and others referring to fig3 - Error, please check;

R2.3 Corrected, thanks.

Line 283 - should link the term "hysteretic effect" to the well-defined term "legacy P" shown in many publications worldwide.

R2.5 We have adapted the sentence, so it now reads "However, no studied country is projected to experience the hysteretic effect under SSP2 due to a lack of sufficient residual (or legacy) phosphorus"

Best regards

Paulo Pavinato

Reviewer 3:

Thank you for making the changes you have made to this manuscript, which is much improved.

R3.1 We are glad you feel the manuscript is much improved and thank you for your time reviewing our work.

The key issue now concerns proofreading and tidying up the English. The comma, in particular, is poorly used and there are written errors and inconsistencies - with the following examples:

1) First sentence of abstract needs to be rewritten

R3.2 The sentence now reads “Sub-Saharan Africa must increase agricultural production more than any other continent by the year 2050. Historically other continents achieved this through intensification methods including the increased use of fertilisers”.

2) Be consistent with the use of calory or calorie as the singular of calories

R3.3 We have corrected this to “calory” throughout, thanks.

3) Line 76-77 what do you mean: ‘for the duration’? This is a completely unspecified term

R3.4 This now reads “to 2050”.

4) Lines 91-96 need careful proof reading

R3.5 This has been proofread with changes made.

5) Line 155: is this correct: “800 % from US\$ 50 to US\$ 50 430”?

R3.6 We have corrected this to “US\$ 430”, thanks.

In terms of the technical content, I appreciate that SSP5 relates to 'fossil fuelled development', but the emphasis is on fossil fuels for energy rather than mined materials. I think you should make it clear that your interpretation of SSP5 specifically encompasses mined raw materials.

R3.7 We added “SSP5 was originally envisaged as “fossil fuelled growth”²³, however, we broaden the term to mean environmentally exploitative (e.g. expanded agricultural lands, high water use, high greenhouse gas emissions) within the context of high GDP growth²²” (*Line 238 – Introduction*).

However, this does lead to your statement in the Conclusions that "economic growth is stimulated by the extractive industries (SSP5)". This aspect is not discussed specifically earlier, and so is a new idea that enters the conclusions. I think it should be discussed, as one of the threads running through the first draft, and still in the second draft, is an apparent lack of understanding of how the mining industry works, and of how the term 'reserve' is very specifically defined in regulations to provide a figure that investors can trust - to a common standard internationally. 'Reserves' are an entirely artificial construct that have no relationship to the rocks in the ground, other than the figure reported has been proved by robust investigation of those rocks. So the reserves figures for many commodities in the USGS reports that you cite are constant from year to year, simply because the pace of their definition equals the pace of their extraction. For an investor, that is a good place to be.

R3.8 We were not clear about the difference between a reserve and a resource. We have added the following paragraph to the introduction to clarify this. We believe the other analysis of setting our estimates within current reserve estimates remains valid.

Line 117 (Introduction) - There are an estimated 65 billion tonnes of phosphate rock in global reserves (i.e. the amount known to be economically extractable at high enough quality for use), however, there are estimated to be more than 300 billion tonnes in global geological resources (i.e. estimate concentrations not necessarily economically extractable) ¹⁵. Uncertainty remains about their grade and accessibility of these resources, however, geological resources can becoming economic reserves with technological and economic changes (e.g. decreasing extraction and processing price and or increased scarcity increasing phosphorus price) ^{10,13}. Currently, peak phosphorus is predicted to occur between 2070 and 2080 ⁹ but there are immediate challenges to phosphorus. The most critical of these is that 85 % of the reserves are in a disputed territory between Morocco and Western Sahara meaning that phosphorus is extremely geopolitically vulnerable to both national monopolies and political instabilities.

The extractive industries do not stimulate economic growth by themselves. They only exist because investors are convinced that there is demand for their products, so it is the other way round: society stimulates demand for extraction. Just look at the current lithium 'gold rush' as an example of how that works as an SSP5 driver designed to counter the fossil fuels. You can see this relationship with investors in the York Potash project, where \$billions have been raised from investors to build a mine on the basis of long-term purchase agreements for the fertiliser minerals produced by the mine once it is built. Those agreements give confidence that the investors will get their money back.

R3.9 Agreed, we did not intend to give this impression. To clarify this point and in response at R3.7 we have described SSP5 has environmentally exploitative with high GDP growth – we wish to avoid causality on this point as it takes us down a tangent. We have made the following change:

Line 473 (Conclusions) - The lowest increases in application rates, 310 %, are likely to be under an increasingly unequal pathway (SSP4) where GDP does not increase and neither does meat and milk production. The highest increase in phosphorus use (620 %) is in the environmentally exploitative SSP5 with its high levels GDP growth (R3.7).

Additional Changes:

1. To ensure the manuscript is inline with the Nature Communications guidelines we have altered the abstract. It now reads:

Sub-Saharan Africa must increase agricultural production dramatically by 2050: phosphate fertilisers are critical to this. We project that between 2020 and 2050 Sub-Saharan African will require between a 310 % and 620 % increase in phosphorus depending on the shared socioeconomic pathway (SSP) taken. The lowest increase is in an unequal society (SSP4) where GDP, crop, meat and milk production are low. The highest increase is in an environmentally costly scenario with expanded agricultural lands but high levels of GDP growth, crop, meat and milk production (SSP5). The sustainable pathway (SSP1) has a high GDP growth for the low expansion of agricultural lands and phosphorus use whilst the middle of the road (SSP2) and increased nationalism pathways (SSP3) are middling but environmentally costly pathways for limited GDP growth. SSP2 requires approximately 440 million tonnes of phosphate rock which is within current reserve estimates but may cost US\$ 50 ± 15 billion for the raw material alone.

2. Throughout the manuscript we have changed “business as usual” to “middle of the road” this brings the manuscript in-line with the IMAGE 3.2 language now used.
3. IMAGE 3.0 has been updated to IMAGE 3.2 which was the version used in this study.

New Figures

Figure 1 Historical and future Sub-Saharan African: (A) population growth (millions of people); (B) GDP (per capita US\$); (C) Meat production (Mg/yr); (D) Milk production (Mg/yr); (E) Crop production (Mg/yr); (F) Yield (Mg dry matter/ha/yr); and (G) Crop production areas (Mha) projected by IMAGE 3.2 for SSP1-5²⁷.

Figure 2 For SSP2, projected phosphate fertiliser price broken down into rock (red), production (blue), transport (green), tax (purple) and margin (orange) costs for (A) Nigeria, (B) Ethiopia, (C) South Africa and (D) Ghana. The percentage values annotated indicate the contribution to the total costs from phosphate rock.

Figure S3 IMAGE 3.2 crop production predictions over the time period 1970 -2050 for SSP1 – 5 divided into different crop types.

REVIEWER COMMENTS

Reviewer #1 (Remarks to the Author):

Comments on Revision 3

1. GENERAL COMMENTS:

The authors have made some good improvements to the previous version. Substantial changes have now been made by the authors in response to the latest review round. These have addressed some of the concerns I had raised (e.g. adding other SSP projections, improving figures/graphs, clarifying the food security context), but not all concerns.

The biggest concern I still have pertains to the economic component assumptions, in particular:

a) the assumption that South Africa doesn't import phosphate because it holds large phosphate rock reserves, when in fact South Africa is the largest importer of phosphate fertilisers in all Sub-Saharan Africa (see below) which has serious implications for your results. Look at the publicly available phosphate trade data (e.g. FAOSTATS).

b) why phosphate rock has been used as the basis for the economic (trade) analysis, when phosphate fertiliser imports are significantly greater than phosphate rock imports. i.e. phosphorus is predominantly imported into Sub-Saharan African countries as MAP, DAP and NPK fertilisers. Phosphate rock imports are often negligible or orders of magnitude smaller. I raised this in previous review rounds. This shows a lack of understanding of the market. Again, check the readily available phosphate trade data.

c) the price of phosphate fertilisers still has some significant gaps, perhaps also indicating a lack of understanding of phosphate markets (See below). While the authors have now added several additional components to the cost (including shipping and tax, which is good), there are still large gaps. Phosphoric acid (not phosphate rock) is the intermediate product used in fertiliser production (i.e. sulphur reacted with phosphate rock). Nitrogen/ammonia (and hence gas) are also a significant cost components in the production of phosphate fertilisers. The FOB market price of phosphate fertilisers are around 300-600% times greater than the market price of phosphate rock.

If the manuscript isn't rejected, I would suggest one of two ways forward regarding the economic component. Either: 1. remove the economic component altogether from the manuscript as it appears to be the weakest part (and potentially flawed), or 2. redo the analysis based on realistic/accurate assumptions for phosphate FERTILISER import/trade (not phosphate ROCK), ensuring these are evidence-based informed assumptions (not presumptions like SA self-sufficiency) and reframe the interpretation of your results.

Take care when referring to 'achieving food security' in the quantitative analysis – this needs to be consistent with your intro definitions of food security, or defined otherwise using more explicit terms (e.g. are you referring to a single aspect of food security, like closing the yield gap, or phosphorus demand being met by supply etc? or also including food access, nutrition etc. If so, be explicit). See some specific examples in the line-by-line comments below.

Further, the new/revised text has introduced many new inconsistencies, ambiguities, and grammatical issues. These are minor in nature, but many in frequency! I have detailed some below in the line-by-line comments, but cannot address everything. The authors should have addressed many of these before re-submitting (e.g. consistency re IMAGE 3 or 3.2? Which is the correct manuscript title?). The manuscript now needs another serious grammar and proof check.

There are some jargon terms which could be clarified to reach a wider audience who might not be familiar with soil phosphorus modelling terms (not essential, but useful if you would like this to be read more widely). I have suggested some of these below in the line-by-line comments.

2. DETAILED COMMENT ON COSTS ASSUMPTIONS:

Further to the general comments above about cost assumptions:

>>>> The market price of fertilisers (e.g DAP) can be 300%-600% times greater than the price of phosphate rock (see World Bank Pink Sheets). E.g. when Phosphate rock was \$74/tonne in Nov 2019, DAP was \$248/tonne. When Phosphate rock was \$96/tonne in March 2021, DAP was \$534/tonne. The market price of phosphate fertilisers includes the price of nitrogen/ammonia, gas, etc, in addition to other production costs. N prices also fluctuate greatly with the price of gas, affecting the price of DAP/MAP/NPK and hence farmer market access. The authors have added 'production costs' to the price of phosphate rock, but only at \$38/tonne, which still leaves a significant shortfall between the price of phosphate rock and the price of fertilisers.

>>>> The farm-gate price of fertilisers can be an additional 2-5 times greater than the market price of fertilisers. The authors have added some of these additional costs to reach the port (e.g. international transportation, taxes etc), and refer to IFDC 2007. But as also noted in IFDC 2007, in-land costs including transportation and dealer margins can be very significant in Africa. Especially for land-locked African countries.

>>>> The authors note that these figures are for illustrative purposes, and not for predictions (i.e. not accounting for geopolitical influences such as those playing out right now – phosphate prices have spiked 400% since early 2020). I think it is fine and appropriate to exclude price spikes due to geopolitical factors as the authors clearly note, although it is worth also mentioning (qualitatively) in a sentence what the consequences of fertiliser price spikes are for African nations (nations with and without fertiliser subsidies and substantial legacy soil phosphorus accumulations that are in accessible pools to farmers).

3. SPECIFIC COMMENTS ON NEW TEXT:

Line numbers refer to those in the clean manuscript (word doc), not the track change version, which has different line numbers:

TITLE:

Which is the latest title? The revision documents have different titles: The track change manuscript is "The impact of phosphorus on projected Sub-Saharan Africa food security futures", while the Response to Reviewers and Supplementary Material refers to "The role of phosphorus in a food secure Sub-Saharan African future" ?

ABSTRACT:

Line 9: Sub-Saharan Africa must improve food security urgently right now (as you note in the intro), in addition to the longer-term (2050 in your case).

Line 11: "budget" – for a wider readership, you might want to say "use in agriculture" or something less jargony.

Line 12: 'can' "achieve food security"?

Line 14 "low phosphorus" what? Low phosphorus use? low phosphorus input?

Line 18: "may cost US\$ 50 ± 15 billion for the raw material alone" – cost Sub-Saharan Africa? Between 2020-2050?

INTRO

Line 1.1. "Rock Phosphorus" – use 'Phosphate rock'.

Line 47 – "are the most yield limiting" – in what? 'in agriculture'? 'in crop production'?

Line 49 – "Today these regions can rely on residual (or legacy)" – add 'in theory' or something to clarify that this isn't widespread practice (but needs to be)

Line 56 – "NPKs" – remove the "s" – should read 'NPK fertilisers'

Line 56/57 – add a space to 'diammonium phosphate' and 'monoammonium phosphate'

Line 62/63 – "instead peak in phosphorus will occur followed by gradual decline in use" – the peak in peak phosphorus is due to increasing demand exceeding supply of high-grade phosphate rock, not a peak/decline in demand (the text reads as if it is the latter). The peak occurs long before all the reserves have been depleted, because it is economically/technically/legally/politically no longer feasible to extract the remaining reserves (same as observed empirically for oil).

Line 73/74 – It might be appropriate to mention that the price of phosphate rock has now spiked 350% again between 2020-2022....

Line 131-135: A reminder that the intensification-extensification spectrum is one option to meet future food demand, but that there are also other measures (outside the scope of the present manuscript), such as reducing food waste in the food value chain, changing diets, etc, which will contribute to closing the gap between food demand and production. i.e. we don't need to achieve such high production if we're not wasting so much and not demanding so much with resource-intensive diets.

Line 146-47: the ratios of crops to one another staying the same, implies the scenarios don't take account of changing diets towards more healthy foods (e.g. more legumes, vegetables, less starchy vegetables).

Lines 154-163: This is a strong paragraph/point.

Lines 165: "the environmental challenges of meeting.." – specifically for SSA?

Line 202: "Reach food security" – how is this defined for the purpose of this paper? Also on line 217 "a food secure future can be achieved". Table 1 indicates increasing or decreasing aspects of food security, but what does '100%' achieving food security look like? Do you mean simply that the phosphorus yield gap is closed? Or that phosphorus demand for food can be met by supply? Or also measures of food access, e.g. that all citizens in SSA can afford to purchase healthy food etc? Really important to be explicit (e.g. 'food demand in SSA can be met') or whatever you mean. And make sure how you define it is consistent with and justified by your intro text on food security.

Figure 2: There are 2 different headings with slightly different meanings. "phosphorus required" implies something different to "phosphorus application". The latter is a result of various SSP scenarios, while 'required' implies P is needed to meet a food or agricultural target?

Line 233: define "elemental phosphorus uptake" for readers (e.g. the amount of phosphorus taken up by (or harvested in?) crops).

Figure 3 (Line 241): B) "phosphorus fertiliser application (kg ha⁻¹)" – does this include manure or other sources of P? Only wondering as Figure 6 does include manure, when comparing P inputs to uptake. If not, please explain somewhere the assumptions behind including it in one part of the analysis and not the other.

Line 250: be specific – "The increases in 'phosphate fertiliser' application"

Line 256: "Many countries including Nigeria, Côte d'Ivoire, Ghana and Ethiopia have imported

almost all of the phosphate rock used” – but they import phosphorus overwhelmingly as fertiliser, not phosphate rock? e.g. Ethiopia (2015) – 700,000 tonnes.

Line 276: caveats – also the significant assumptions that a) these countries import P as phosphate rock (which they mostly don't), and b) that South Africa doesn't import P because it produces so much phosphate rock, which it is the largest importer of fertilisers in SSA (e.g. see 2015 according to FAOSTATS).

Line 317” “except South Africa which is assumed to be self-producing” – are you just assuming this based on the fact that they have extremely large phosphate rock reserves? South Africa also imports phosphate fertilisers – the largest importing country in SSA, according to FAOSTATS data.

Line 355: “achieve food security” – again, clarify what you actually mean here. If you mean ‘meet future food demand’, then say this as it alone does not equate to ‘food security’.

CONCLUSION: Many others have addressed/analysed phosphorus fertiliser demand, yield gaps, fertiliser prices, in Sub Saharan Africa (some in the context of the Shared Socioeconomic Pathways). It would be good to have a brief discussion to contextualise your results/findings in relation to the existing literature. How do your results compare or add to these? (e.g. as mentioned in the previous review round – how does this compare/relate to Springmann et al findings, or those below (some of which you reference in the article, but don't explicitly compare your findings to).

Phosphorus for Sustainable Development Goal target of doubling smallholder productivity
C. Langhans, A. H. W. Beusen, J. M. Mogollón & A. F. Bouwman
Nature Sustainability volume 5, pages57–63 (2022)
<https://www.nature.com/articles/s41893-021-00794-4>

Future agricultural phosphorus demand according to the shared socioeconomic pathways
Mogollón, J. M., Beusen, A. H. W., van Grinsven, H. J. M., Westhoek, H. & Bouwman, A. F..
Glob. Environ. Change 50, 149–163 (2018).

Fertilizer and grain prices constrain food production in sub-Saharan Africa
Camila Bonilla-Cedrez, Jordan Chamberlin & Robert J. Hijmans
<https://www.nature.com/articles/s43016-021-00370-1>

Beyond fertilizer for closing yield gaps in sub-Saharan Africa
André F. Van Rooyen, Henning Bjornlund & Jamie Pittock
<https://www.nature.com/articles/s43016-021-00386-7>

Supplementary Material: Line 12: Are you using IMAGE 3.0 or 3.2? Be consistent.

RESPONSE TO REVIEWER COMMENTS

Reviewer #1 (Remarks to the Author):

Comments on Revision 3

1. GENERAL COMMENTS:

The authors have made some good improvements to the previous version. Substantial changes have now been made by the authors in response to the latest review round. These have addressed some of the concerns I had raised (e.g. adding other SSP projections, improving figures/graphs, clarifying the food security context), but not all concerns.

The biggest concern I still have pertains to the economic component assumptions, in particular:

a) the assumption that South Africa doesn't import phosphate because it holds large phosphate rock reserves, when in fact South Africa is the largest importer of phosphate fertilisers in all Sub-Saharan Africa (see below) which has serious implications for your results. Look at the publicly available phosphate trade data (e.g. FAOSTATS).

b) why phosphate rock has been used as the basis for the economic (trade) analysis, when phosphate fertiliser imports are significantly greater than phosphate rock imports. i.e. phosphorus is predominantly imported into Sub-Saharan African countries as MAP, DAP and NPK fertilisers. Phosphate rock imports are often negligible or orders of magnitude smaller. I raised this in previous review rounds. This shows a lack of understanding of the market. Again, check the readily available phosphate trade data.

c) the price of phosphate fertilisers still has some significant gaps, perhaps also indicating a lack of understanding of phosphate markets (See below). While the authors have now added several additional components to the cost (including shipping and tax, which is good), there are still large gaps. Phosphoric acid (not phosphate rock) is the intermediate product used in fertiliser production (i.e. sulphur reacted with phosphate rock). Nitrogen/ammonia (and hence gas) are also a significant cost components in the production of phosphate fertilisers. The FOB market price of phosphate fertilisers are around 300-600% times greater than the market price of phosphate rock.

If the manuscript isn't rejected, I would suggest one of two ways forward regarding the economic component. Either: 1. remove the economic component altogether from the manuscript as it appears to be the weakest part (and potentially flawed), or 2. redo the analysis based on realistic/accurate assumptions for phosphate FERTILISER import/trade (not phosphate ROCK), ensuring these are evidence-based informed assumptions (not presumptions like SA self-sufficiency) and reframe the interpretation of your results.

P1) We are grateful for the opportunity to respond. We have elected to remove most of the economic component including figure 5 and the related text. Moving into the economics of phosphorus markets and imports/ exports was outside of our area of expertise and took the paper down an unnecessary tangent. However, the original purpose of providing some costs was to contextualise the findings and we still think this is important.

Most of the context is food security from the van Meijl data but we will still provide bulk indicative costs at continental level and country level for context but now do this of the DAP fertiliser price rather than the phosphate rock price. We make no assumptions on trade, taxation or transportation instead we simply state that this would be the market price of DAP and provide the caveats (including farmgate caveats) that this is an illustrative contextualising calculation.

Line 306 - Our results demonstrate that the minimum total projected Sub-Saharan African phosphate rock requirement between 2020 and 2050 is 440 million tonnes (**Error! Reference source not found.A**) which represents less than 1 % of current known global reserves (65 billion tonnes¹⁹). These results, however, assume that the system of phosphate rock mining to phosphorus application onto arable soils is 100 % efficient, whereas a more accurate figure is likely to be 80 %. It is unknown how efficiency will change over the next three decades⁴⁸. Therefore, our phosphate rock estimate represents a minimum amount of phosphate required.

The increases in phosphate application are not evenly distributed. From a baseline of 2010 to 2020, we projected that phosphate requirement in Nigeria, Ethiopia, South Africa and Ghana will increase by 934 %, 375 %, 148 % and 130 %, respectively, over the 30 years to 2050. By 2050, under middle of the road SSP2, we project this consumption to be 3.19, 2.05, 0.64 and 0.92 million tonnes of fertiliser, respectively, if all is applied as DAP. For comparison, our modelled results from between the years 2010 and 2020 show that Nigeria, Ethiopia, South Africa and Ghana would have consumed an average of 0.29 ± 0.22 , 0.45 ± 0.14 , 0.44 ± 0.04 and 0.05 ± 0.08 million tonnes per year of fertiliser if all was applied as DAP (**Error! Reference source not found.B** and C). Were all applied as TSP the mass would be approximately 105 % of these values whilst if they were applied as MAP the mass would be between 68% and 95 % of the values depending on P_2O_5 concentration^{36,49,50}.

All countries we studied are required to increase their phosphate fertiliser market price expenditure. We project the total phosphate fertiliser market price cost to Sub-Saharan Africa between 2020 and 2050 to be US\$ 130 ± 25 billion were all of this to be applied as DAP. For the four highest consuming countries, Nigeria, Ethiopia, South Africa and Ghana the market price expenditure in the year 2050 is projected to be US\$ 1300 ± 230 , US\$ 815 ± 150 , US\$ 250 ± 45 and US\$ 370 ± 70 million per year, respectively, up from a mean of US\$ 105 ± 70 , US\$ 180 ± 50 , US\$ 180 ± 30 and US\$ 20 ± 25 million US\$ per year, respectively, between 2010 and 2020 (**Error! Reference source not found.B**). Were all of this applied as TSP the total cost would be approximately 90 % of the DAP price given TSP has a lower price but also a lower P_2O_5 concentration²¹.

To provide context to these figures we assessed phosphate fertiliser (as DAP) expenditure as a percentage of independently projected real GDP for each country (**Error! Reference source not found.**)⁵¹. Real GDP adjusts for inflation of goods and services by an economy for a given year and is expressed at base year prices. The results from South Africa and Ethiopia show that both countries have had historical peak expenditures in 1975 and 2008, respectively, but that this expenditure decreased and will continue to do so until 2050. On the other hand, we predict that Nigeria and Ghana have yet to reach their peak expenditure on fertiliser and will experience this peak around the year 2030 with Ghana experiencing a considerably higher peak than Nigeria (**Error! Reference source not found.D**).

Whilst these estimates provide insightful information about the phosphorus fertiliser use and cost in Sub-Saharan Africa in the coming decades, they should be considered approximations and have important caveats. The first caveat is that in practice a range of fertilisers exist, and much phosphate will be added as part of NPK fertilisers. We justify the choice of DAP since it is a widely used phosphate fertiliser and NPK has a variety of compositions^{36,49,50} which prohibits the application of a single value required at the scale we are working. The second caveat is that this is not a calculation of the farmgate price, which usually exceeds the fertiliser price due to additional costs and can be between two to five times greater than the market price of fertilisers⁵⁰. The final caveat is that our projections assume that the phosphate fertiliser stays within the price limits of the 2010 to 2020 decade (US\$ 405 ± 70 per tonne of DAP²¹). This represents the decade of highest sustained and most fluctuating prices but, at the time of writing, the

price is spiking almost twice this value. Nevertheless, as with previous spikes we expect this value to come down in the long term ²¹. The extent to which farmers can maintain high yields can overcome short term fertiliser price spikes by using less or no fertiliser is partly dictated by the amount of residual (or legacy) phosphorus and the so-called hysteretic crop uptake effect.

Take care when referring to 'achieving food security' in the quantitative analysis – this needs to be consistent with your intro definitions of food security, or defined otherwise using more explicit terms (e.g. are you referring to a single aspect of food security, like closing the yield gap, or phosphorus demand being met by supply etc? or also including food access, nutrition etc. If so, be explicit). See some specific examples in the line-by-line comments below.

P2) Good point we will use relative terms (e.g. "Becomes more food secure" and "becomes less food secure"). We also periodically remind the reader that this is within the context of "price, availability and risk of hunger".

Further, the new/revised text has introduced many new inconsistencies, ambiguities, and grammatical issues. These are minor in nature, but many in frequency! I have detailed some below in the line-by-line comments, but cannot address everything. The authors should have addressed many of these before re-submitting (e.g. consistency re IMAGE 3 or 3.2? Which is the correct manuscript title?). The manuscript now needs another serious grammar and proof check.

P3) We apologise for this. We have addressed all of the reviewer's line by line comments and three authors have given a thorough proof read. We do not list every corrected typo but these are clearly visible on the track changes document.

There are some jargon terms which could be clarified to reach a wider audience who might not be familiar with soil phosphorus modelling terms (not essential, but useful if you would like this to be read more widely). I have suggested some of these below in the line-by-line comments.

P4) We have tried to remove jargon where we felt they were still present in particular we have focussed on reminding the reader each time what the different SSPs mean (e.g. sustainable SSP1).

2. DETAILED COMMENT ON COSTS ASSUMPTIONS:

Further to the general comments above about cost assumptions:

>>>> The market price of fertilisers (e.g DAP) can be 300%-600% times greater than the price of phosphate rock (see World Bank Pink Sheets). E.g. when Phosphate rock was \$74/tonne in Nov 2019, DAP was \$248/tonne. When Phosphate rock was \$96/tonne in March 2021, DAP was \$534/tonne. The market price of phosphate fertilisers includes the price of nitrogen/ammonia, gas, etc, in addition to other production costs. N prices also fluctuate greatly with the price of gas, affecting the price of DAP/MAP/NPK and hence farmer market access. The authors have added 'production costs' to the price of phosphate rock, but only at \$38/tonne, which still leaves a significant shortfall between the price of phosphate rock and the price of fertilisers.

P5) We believe the approach outlined at P1 address all these concerns.

>>>> The farm-gate price of fertilisers can be an additional 2-5 times greater than the market price of fertilisers. The authors have added some of these additional costs to reach the port (e.g. international transportation, taxes etc), and refer to IFDC 2007. But as also noted in IFDC 2007, in-land costs including transportation and dealer margins can be very significant in Africa. Especially for land-locked African countries.

P5) We believe the approach outlined at P1 address most of these concerns and additionally we explain how farmgate price is 2 to 5 times higher to contextualise the DAP price.

>>>> The authors note that these figures are for illustrative purposes, and not for predictions (i.e. not accounting for geopolitical influences such as those playing out right now – phosphate prices have spiked 400% since early 2020). I think it is fine and appropriate to exclude price spikes due to geopolitical factors as the authors clearly note, although it is worth also mentioning (qualitatively) in a sentence what the consequences of fertiliser price spikes are for African nations (nations with and without fertiliser subsidies and substantial legacy soil phosphorus accumulations that are in accessible pools to farmers).

We have added two sentences to address this:

Line 96 - Recently, the phosphate price has peaked again increasing 250 % in the year from April 2021 to April 2022.

Line 368 - “at the time of writing, the price is spiking almost twice this value.”

Line 394 - “The extent to which farmers can maintain high yields can overcome short term fertiliser price spikes by using less or no fertiliser is partly dictated by the amount of residual (or legacy) phosphorus and the so-called hysteretic crop uptake effect.

3. SPECIFIC COMMENTS ON NEW TEXT:

Line numbers refer to those in the clean manuscript (word doc), not the track change version, which has different line numbers:

TITLE:

Which is the latest title? The revision documents have different titles: The track change manuscript is “The impact of phosphorus on projected Sub-Saharan Africa food security futures”, while the Response to Reviewers and Supplementary Material refers to “The role of phosphorus in a food secure Sub-Saharan African future” ?

Supplementary material has been corrected.

ABSTRACT:

Line 9: Sub-Saharan Africa must improve food security urgently right now (as you note in the intro), in addition to the longer-term (2050 in your case).

Changed to “urgently improve”.

Line 11: “budget” – for a wider readership, you might want to say “use in agriculture” or something less jargony.

Changed to “use in agriculture”.

Line 12: ‘can’ “achieve food security”?

Corrected

Line 14” “low phosphorus” what? Low phosphorus use? low phosphorus input?

Corrected to “use”.

Line 18: “may cost US\$ 50 ± 15 billion for the raw material alone” – cost Sub-Saharan Africa? Between 2020-2050?

Changed to: (Line 20) “Whilst this is within the current global reserve estimates the market price alone for a commonly used fertiliser (DAP) would cost US\$ 130 ± 25 billion for agriculture over the period 2020 to 2050 and the farmgate price could be two to five times higher due to additional costs (e.g. transport, taxation etc.).

INTRO

Line 1.1. “Rock Phosphorus” – use ‘Phosphate rock’.

Corrected.

Line 47 – “are the most yield limiting” – in what? ‘in agriculture’? ‘in crop production’?

Added “for plant growth”.

Line 49 – “Today these regions can rely on residual (or legacy)” – add ‘in theory’ or something to clarify that this isn’t widespread practice (but needs to be)

Corrected

Line 56 – “NPKs” – remove the “s” – should read ‘NPK fertilisers’

Corrected

Line 56/57 – add a space to ‘diammonium phosphate’ and ‘monoammonium phosphate’

Corrected

Line 62/63 – “instead peak in phosphorus will occur followed by gradual decline in use” – the peak in peak phosphorus is due to increasing demand exceeding supply of high-grade phosphate rock, not a peak/decline in demand (the text reads as if it is the latter). The peak occurs long before all the reserves have been depleted, because it is economically/technically/legally/politically no longer feasible to extract the remaining reserves (same as observed empirically for oil).

Corrected to: (Line 72) But despite concerns in recent decades ^{16,17} humanity is unlikely to run out of phosphate rock and instead peak phosphorus will be reached at the point at which demand exceeds the supply of high grade phosphate (analogous to the more well-known “peak oil” concept).

Line 73/74 – It might be appropriate to mention that the price of phosphate rock has now spiked 350% again between 2020-2022....

Good point. We added: (Line 96) Recently, the phosphate rock price has peaked again increasing 250 % in the year from April 2021 to April 2022²⁰.

Line 131-135: A reminder that the intensification-extensification spectrum is one option to meet future food demand, but that there are also other measures (outside the scope of the present manuscript), such as reducing food waste in the food value chain, changing diets, etc, which will contribute to closing the gap between food demand and production. i.e. we don’t need to achieve such high production if we’re not wasting so much and not demanding so much with resource-intensive diets.

We have added (line 46): Food waste also affects access to food: the UN estimates 930 million tons of food are wasted each year and minimizing food waste through improved storage and distribution would reduce some production requirements ⁷.

Line 146-47: the ratios of crops to one another staying the same, implies the scenarios don’t take account of changing diets towards more healthy foods (e.g. more legumes, vegetables, less starchy vegetables).

This is the input data from IMAGE 3.2 rather than our output data and should remain consistent with van Meijl's food security paper for comparison.

Lines 154-163: This is a strong paragraph/point.

Thanks.

Lines 165: "the environmental challenges of meeting.." – specifically for SSA?

Corrected.

Line 202: "Reach food security" – how is this defined for the purpose of this paper? Also on line 217 "a food secure future can be achieved". Table 1 indicates increasing or decreasing aspects of food security, but what does '100%' achieving food security look like? Do you mean simply that the phosphorus yield gap is closed? Or that phosphorus demand for food can be met by supply? Or also measures of food access, e.g. that all citizens in SSA can afford to purchase healthy food etc? Really important to be explicit (e.g. 'food demand in SSA can be met') or whatever you mean. And make sure how you define it is consistent with and justified by your intro text on food security.

Our changes at P2 have addressed this throughout the manuscript.

Figure 2: There are 2 different headings with slightly different meanings. "phosphorus required" implies something different to "phosphorus application". The latter is a result of various SSP scenarios, while 'required' implies P is needed to meet a food or agricultural target?

Corrected to required. Changed in the caption.

Line 233: define "elemental phosphorus uptake" for readers (e.g. the amount of phosphorus taken up by (or harvested in?) crops).

A definition has been added to the definitions section and the introduction.

Figure 3 (Line 241): B "phosphorus fertiliser application (kg ha⁻¹)" – does this include manure or other sources of P? Only wondering as Figure 6 does include manure, when comparing P inputs to uptake. If not, please explain somewhere the assumptions behind including it in one part of the analysis and not the other.

To assess this effect, total applied phosphorus which, unlike the rest of this manuscript includes manure, is compared to phosphorus uptake. Total phosphorus is used since this is a physio-chemical process and both components will effect uptake.

Line 250: be specific – "The increases in 'phosphate fertiliser' application"

Section removed.

Line 256: "Many countries including Nigeria, Côte d'Ivoire, Ghana and Ethiopia have imported almost all of the phosphate rock used" – but they import phosphorus overwhelmingly as fertiliser, not phosphate rock? e.g. Ethiopia (2015) – 700,000 tonnes.

Section removed.

Line 276: caveats – also the significant assumptions that a) these countries import P as phosphate rock (which they mostly don't), and b) that South Africa doesn't import P because it produces so much phosphate rock, which it is the largest importer of fertilisers in SSA (e.g. see 2015 according to FAOSTATS).

Section removed.

Line 317” “except South Africa which is assumed to be self-producing” – are you just assuming this based on the fact that they have extremely large phosphate rock reserves? South Africa also imports phosphate fertilisers – the largest importing country in SSA, according to FAOSTATS data.

Section removed.

Line 355: “achieve food security” – again, clarify what you actually mean here. If you mean ‘meet future food demand’, then say this as it alone does not equate to ‘food security’.

Conclusion now starts: (**Line 452**) For Sub-Saharan Africa, two pathways – sustainable SSP1 and fossil fuelled SSP5 – increase both food security (in terms of price, availability, and risk of hunger) and GDP by 2050..

CONCLUSION: Many others have addressed/analysed phosphorus fertiliser demand, yield gaps, fertiliser prices, in Sub Saharan Africa (some in the context of the Shared Socioeconomic Pathways). It would be good to have a brief discussion to contextualise your results/findings in relation to the existing literature. How do your results compare or add to these? (e.g. as mentioned in the previous review round – how does this compare/relate to Springmann et al findings, or those below (some of which you reference in the article, but don’t explicitly compare your findings to).

On this point we disagree with the reviewer. We believe the conclusions as they stand are strong and provide a clear message from this paper. The findings are contextualised against the van Meijl findings which we believe to be the most appropriate for our study given they also used the SSPs. We have tried re-writing the conclusions in multiple ways including using the Springmann et al. findings but were not satisfied with them. We believe the best place for Springmann is in the introduction to frame the question as we have done and to use van Meijl SSPs to contextualise our results.

Additional References

Phosphorus for Sustainable Development Goal target of doubling smallholder productivity

C. Langhans, A. H. W. Beusen, J. M. Mogollón & A. F. Bouwman

Nature Sustainability volume 5, pages57–63 (2022)

<https://www.nature.com/articles/s41893-021-00794-4>

Line 196 - Phosphorus availability is a serious limiting factor in achieving the second SDG in tropical countries ³⁸.

Future agricultural phosphorus demand according to the shared socioeconomic pathways

Mogollón, J. M., Beusen, A. H. W., van Grinsven, H. J. M., Westhoek, H. & Bouwman, A. F..

Glob. Environ. Change 50, 149–163 (2018).

Line 200 - The amount of phosphorus supplied from fertiliser varies across the continent: in 2005 52 % of phosphorus in North Africa came from fertiliser, 29 % in East Africa and 16 % in Western Africa

Line 476 - GDPPS uses a theory driven geochemical fitting to overcome limitations induced in the conventional DPPS model where soils are highly weather such as tropical locations ³⁹.

Fertilizer and grain prices constrain food production in sub-Saharan Africa

Camila Bonilla-Cedrez, Jordan Chamberlin & Robert J. Hijmans

<https://www.nature.com/articles/s43016-021-00370-1>

Line 167 - In Sub-Saharan Africa the highest potential yields are in East Africa from Ethiopia to Zimbabwe and West Africa from Nigeria to Sierra Leone but many of these areas lack potential profitability ³⁶. The lack of potential profitability means there is a lack of financial incentives at the farm scale to close the so-called economic yield gap. The economic yield gap is the difference between current

profit and maximum potential profit given the yield and is currently a quarter of the ecological yield gap³⁶.

Beyond fertilizer for closing yield gaps in sub-Saharan Africa

André F. Van Rooyen, Henning Bjornlund & Jamie Pittock

<https://www.nature.com/articles/s43016-021-00386-7>

We haven't included this reference since it is a summary of Bonilla-Cedrez paper which we feel is more appropriate to cite.

Additionally, we have added the following paper which came out last week and the accompanying text:

Line 164 - Where nutrients are very low so-called “ecological intensification” can improve the ecological yield gap. Here, both organic matter is added to the soil and crop diversity is increased – including adding fertility crops to the rotation with³⁵.

MacLaren, C. *et al.* Long-term evidence for ecological intensification as a pathway to sustainable agriculture. *Nat. Sustain.* 2022 1–10 (2022). doi:10.1038/s41893-022-00911-x

Supplementary material

Supplementary Material: Line 12: Are you using IMAGE 3.0 or 3.2? Be consistent.

Corrected to 3.2.

REVIEWERS' COMMENTS

Reviewer #1 (Remarks to the Author):

Comments on Revision 4

This revision reads much better. Stronger, more grounded and clearer to read.

I have no further comments of significance – except the below minor query on Fig 4A. Thanks for your patience on this long journey of reviews and revisions. If published, I hope this article will be received well.

Figure 4A Y-axis units – “Total phosphorus required (1000 tonnes)” – is this meant to be “million tonnes” instead of “1000 tonnes”, to yield the 440 million tonnes of phosphate rock in line 282?